# Hugging Carbon: Quantifying the Training Carbon Emissions of AI Models at Scale

**Xinlei Wang** [1 2]   **Ruibo Ming** [1]   **Jing Qiu** [2]   **Junhua Zhao** [3 4]   **Jinjin Gu** [1]

## Abstract

The scaling-law era has transformed artificial intelligence (AI) from research into a global industry, but its rapid growth also raises concerns over energy usage, carbon emissions, and environmental sustainability. Unlike traditional sectors, the AI industry still lacks systematic carbon accounting methods that support large-scale estimates without reproducing the original training process. This leaves open questions about how large the problem is today and how large it might be in the near future. Given its central role in hosting open-source AI models, the Hugging Face (HF) platform provides a large-scale and publicly accessible corpus for carbon accounting. We estimate aggregate training emissions of HF open-source models using available emissions, energy, compute, and model metadata. To address uneven disclosure quality, we introduce a tiered approach to handle incomplete metadata, supported by empirical regressions that assess estimation reliability. We further introduce AI training carbon intensity (ATCI, emissions per compute), a metric to assess the sustainability efficiency of model training. Our results show that training the most popular open-source models (with over 5,000 downloads) has already resulted in approximately $6.0 \times 10^4$ metric tons of carbon emissions. Overall, this paper provides a scalable, empirically grounded framework for estimating training emissions from incomplete disclosures and informing future carbon reporting standards in the AI industry. Data and code are available at https://github.com/insait-institute/HuggingCarbon.

## 1. Introduction

In the scaling-law era, artificial intelligence (AI) has expanded from academic research into an industry worth hundreds of billions of dollars today, and is projected to reach several trillion dollars by 2030 (UNCTAD, 2025). Large models, spanning computer vision (CV) and large language models (LLMs), are now deployed across critical fields such as the Internet, robotics, energy, and other industrial sectors. This rapid scaling of model size, data, and parameters is driving unprecedented demands for energy (IEA, 2024; Strubell et al., 2019), water (Li et al., 2025; Morrison et al., 2025), and materials (Lee et al., 2025; Bender et al., 2021). Concerns over AI's environmental sustainability are intensifying (Schwartz et al., 2020; Wu et al., 2022; Bashir et al., 2024), as rising emissions risk accelerating climate change and resource strain.

However, these concerns often remain conceptual. While policymakers and researchers broadly acknowledge the challenge, there is still a lack of systematic estimates to the questions of **"how large is the problem today"** and **"how large might it be in the near future"**. In contrast, traditional industries, such as manufacturing and agriculture, already follow established methodologies (Eggleston et al., 2006; Krey et al., 2014) and disclosure standards (International Organization for Standardization, 2018) for product-level life-cycle footprints (Bhatia et al., 2011) as well as industry-wide carbon accounting (Bashmakov et al., 2023). AI, despite its widely recognized environmental implications, still lacks consistent reporting and scalable methodologies for estimating training or inference emissions across a wide range of model families and modalities. Comprehensive and long-term disclosure of the environmental costs of model development and deployment remains limited, and the quality of existing disclosures is often inadequate. This gap makes even a basic understanding of AI's current environmental impacts a pressing and unresolved challenge.

Here, we make a further attempt to bridge these gaps. Unlike previous studies that focused primarily on the carbon footprint of individual models (Strubell et al., 2019; Morrison et al., 2025), we aim to provide a broader, industry-scale perspective on AI's emissions by offering a conceptual estimate of its overall impact. As a lens for this investigation,

[1]INSAIT, Sofia University "St. Kliment Ohridski", Bulgaria [2]University of Sydney, Australia [3]The Chinese University of Hong Kong, Shenzhen [4]Shenzhen Institute of Artificial Intelligence and Robotics for Society, AIRS. Correspondence to: Xinlei Wang <xinlei.wang@insait.ai>, Jinjin Gu <jinjin.gu@insait.ai>.

*Proceedings of the $43^{rd}$ International Conference on Machine Learning*, Seoul, South Korea. PMLR 306, 2026. Copyright 2026 by the author(s).

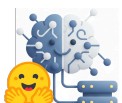 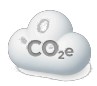 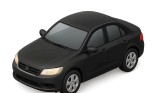 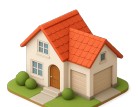 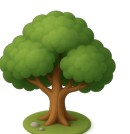 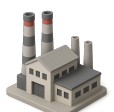

| **5,227 Models**
(>5,000 downloads)
*about 2-4K popular HF models added per year* | **≈60,000**
metric tons CO₂e | **≈42,000 cars**
(annual emissions)
*≈1.6% of passenger-car annual emissions in a mid-sized EU country* | **≈6,000 EU residents**
(annual GHG footprint)
*≈0.2 % of population in a mid-sized EU country* | **≈3.3 million trees**
(one year of absorption) | **≈1 small cement plant**
(half-year emissions) |

*Figure 1.* Estimated training emissions from 5,227 Hugging Face models, compared with equivalent real-world scales (cars (European Environment Agency, 2024), EU residents (Eurostat, 2025), cement plants (IEA, 2025), and trees (Franklin Jr & Pindyck, 2024); tree absorption = 18 kg $CO_2$/tree/year).

we examine open-source models hosted on Hugging Face (HF), the most widely used repository and distribution platform for AI models. The models available on HF represent a substantial share of the open-source community's collective efforts, making them a valuable proxy for estimating emissions in practice. By accounting for the training emissions of these models, we seek to shed light on how much emissions AI training has already emitted and how much emissions its continued scaling may generate. Given the limited quality and scope of existing disclosures, our goal is not to produce exact point estimates but to develop a framework that enables large-scale estimation, empirical validation, and transparent uncertainty analysis. We hope this can offer a bigger-picture view of AI's environmental impact, both at the model level and across the industry.

Accounting for the training emissions of HF models is far from straightforward, requiring a practical methodology. Although open-source models provide a relatively transparent basis for analysis, the information required for carbon accounting remains unevenly disclosed, with many fields requiring manual completion or inference. Meanwhile, reproducing the training process for millions of models would be both infeasible and environmentally wasteful. To address this, we introduce a tiered carbon accounting framework that uses available emissions, energy, compute, and model metadata to estimate training emissions under incomplete disclosure. The key idea is to prioritize directly disclosed emissions or energy information when available, while using compute-based estimation to fill disclosure gaps. For models requiring compute-based estimation, we approximate the total training compute in FLOPs while accounting for differences in network architectures, modalities, and data types. FLOPs are then translated into electricity use through hardware efficiency, power consumption, and runtime amplification factors that capture system-level overheads. Finally, electricity use is converted into emissions using region-specific carbon intensity. This process also defines AI training carbon intensity (ATCI), measured as training emissions per unit of compute, to compare the sustainability efficiency of model training.

In practice, we begin by focusing on models with high download counts and wide adoption, as they not only have substantial ecosystem impact but are also more likely to provide at least partial training-related disclosures. These models cover major research domains such as audio, computer vision (CV), natural language processing (NLP), and multi-modal (MM) learning. Based on the completeness of disclosed information, we classify these models into three tiers: Tier 1 models provide sufficiently detailed information to enable emissions estimates to be cross-checked from multiple perspectives; Tier 2 models disclose only part of the required information, so we estimate the missing components based on empirical patterns observed in Tier 1 models; Tier 3 models provide very limited or no usable disclosure, requiring empirical assumptions for rough estimation. This tiered categorization enables our framework to remain systematic and applicable despite substantial heterogeneity in disclosure practices.

Our estimates suggest that training the 5,227 models with more than 5,000 downloads produced approximately **60,000 metric tons of $CO_2$e** with an uncertainty of $\pm 2.4 \times 10^4$ t$CO_2$e. As shown in Figure 1, the total carbon footprint is comparable to about 1.6% of the passenger-car annual emissions of a mid-sized European country (e.g., Croatia), or the amount of carbon emissions that would require approximately 3.3 million trees to absorb over one year. Since the accounting boundary is defined around open-source models hosted on Hugging Face, the estimate should be interpreted as a conservative lower bound for publicly released open models rather than the full AI industry. The number of HF models continues to rise as thousands of new models are released each year, underscoring their non-negligible emission scale within the open model ecosystem.

## 2. Related Work

**Sustainability of AI.** AI sustainability requires quantifying and mitigating the environmental costs of developing and deploying AI models. Early awareness came from work on energy and policy considerations in deep learning: Strubell et al. (2019) quantified the emissions of training large neural networks and argued that computing should be treated as a scarce resource, while Schwartz et al. (2020)

proposed the "Green AI" agenda, calling for efficiency and environmental impact to be considered alongside accuracy. Patterson et al. (2021) later estimated emissions from models such as GPT-3, showing how data-center efficiency and energy mix affect outcomes.

Subsequent research broadened the scope beyond individual case studies. Wu et al. (2022) surveyed the environmental impacts of AI across data, algorithms, and hardware. Dodge et al. (2022) and Lannelongue & Inouye (2023) introduced location- and time-specific carbon intensity metrics. Case studies such as BLOOM incorporated embodied emissions from hardware manufacturing (Luccioni et al., 2023), while open reports like Llama-2 (Touvron et al., 2023) and OLMo (Groeneveld et al., 2024) disclosed approximate training footprints, providing transparency for reproducible energy studies. In parallel, a range of tools emerged to improve accounting. The ML $CO_2$ Impact Calculator required manual input (Lacoste et al., 2019), CodeCarbon extended this by embedding real-time monitoring into training workflows (Courty et al., 2024), CarbonTracker predicted emissions from early profiling (Anthony et al., 2020), Eco2AI integrated monitoring with PyTorch/TF (Budennyy et al., 2022), and Tracarbon covered device's energy usage (Florian Valeye, 2021). While these tools increased transparency, they remain limited by narrow system boundaries, incomplete hardware coverage, and reliance on average rather than spatiotemporal grid factors.

Recent work has examined downstream deployment, including inference costs (Samsi et al., 2023; Luccioni et al., 2024), fine-tuning trade-offs (Wang et al., 2023), emissions from generative AI usage in HCI research (Inie et al., 2025), and system-level accounting frameworks such as CarbonConnect (Lee et al., 2024). Other studies evaluated optimisation strategies (Fernandez et al., 2025), lifecycle impacts (Morrison et al., 2025), and called for stronger disclosure and policy integration (Luccioni et al., 2025a;b). While these efforts advanced discussions on efficiency, transparency, and governance, they largely address single models or isolated lifecycle stages. The broader ecosystem-level impact remains underexplored. In this paper, we move beyond case studies to systematically estimate the training emissions of thousands of models on Hugging Face, providing a platform-scale perspective on AI's carbon footprint and a baseline for tracking its future trajectory.

**Carbon Accounting.** It refers to the systematic quantification and reporting of greenhouse gas (GHG) emissions, providing reliable foundations for climate policy and sustainability research. The Intergovernmental Panel on Climate Change (IPCC) established a comprehensive methodological framework in the 2006 Guidelines for National Greenhouse Gas Inventories, which has been adopted by countries for sectoral inventories covering energy, industry, and agri-culture (Eggleston et al., 2006). Within this framework, carbon accounting can be differentiated into industry-level accounting, which estimates total emissions from entire sectors throughout production, operation, and supply chains (Bashmakov et al., 2023; United States Environmental Protection Agency, 2025), and product-level accounting, which applies life-cycle assessment (LCA) to a single product or service across its full cradle-to-grave stages (Pankaj Bhatia, Cynthia Cummis, Laura Draucker, David Rich, Holly Lahd and Andrea Brown (WBCSD), 2011; International Organization for Standardization, 2018; Together for Sustainability (TfS) Initiative, 2024). Despite mature practices in other domains, few standardized frameworks exist for carbon accounting of the AI sector. The Software Carbon Intensity (SCI) (Green Software Foundation, 2024) published by the Green Software Foundation (GSF) defines a methodology for carbon accounting of a software system. It only measures the carbon intensity of a software application per functional unit, without using architecture-specific FLOPs or training metadata. Neither IPCC guidelines nor LCA standards extend to AI training or inference, and disclosure is largely absent. Recent steps, such as the EU AI Act, the Energy Efficiency Directive, California's AB 222, and ongoing ISO/IEC drafts (European Union, 2023; 2024; California State Assembly, 2025; ISO, 2025), signal progress, but the AI industry remains insufficiently represented in existing carbon accounting regimes.

**Emissions from AI Training.** Recent studies have estimated the electricity use and carbon emissions of training large models, but typically focus on a few representative cases, leaving ecosystem-level impacts unclear. They have examined training emissions but treated FLOPs as a fixed computational quantity, rather than as part of the core indicator for evaluating carbon efficiency. Strubell et al. (2019) calculate training emissions using measured/reproduced electricity × regional EF for several NLP models (GPT-2, BERT, etc.). Patterson et al. (2021) estimate FLOPs for Google models (T5, Meena, etc.), but emissions are still derived from measured electricity × regional EF, not FLOPs-based estimation. Anthony et al. (2020) and Lacoste et al. (2019) use FLOPs as a proxy for electricity consumption, without analyzing emissions-per-FLOP or cross-model carbon intensity. They consider hardware efficiency (FLOP/s), but none treat FLOPs as part of a standardized or comparable metric (e.g., Emission/FLOP) for model-level sustainability efficiency. Luccioni et al. (2023) compute BLOOM's emissions from internal energy logs and regional emission factors. LLMCarbon (Faiz et al., 2024) infers energy use during training from FLOPs, hardware, and parallelism configurations, and validates its model on a small set of LLMs. However, prior work focuses on single-model or single-architecture case studies (Strubell et al., 2019; Luccioni et al., 2023; Wang et al., 2023; Morrison et al., 2025), de-

pends on complete metadata or internal telemetry (Patterson et al., 2021), or provides experiment-level monitoring tools (Lacoste et al., 2019). They face challenges to scale thousands of models and enable scalable estimations. The key bottleneck, overlooked in prior work, lies in estimating FLOPs, hardware, regions, PUE, and runtime for thousands of heterogeneous models with missing disclosures.

Complementary tools exist: Hugging Face introduced a `co2_eq_emissions` field in 2022 (covering less than 0.2% of repositories). This field relies on CodeCarbon (Courty et al., 2024), which requires detailed runtime logging of hardware power and grid intensity. CarbonTracker (Anthony et al., 2020) monitors real-time CPU/GPU power draw during training and estimates the emissions based on the local grid intensity. It requires full runtime access, hardware telemetry, and controlled training environments that are rarely available from public model metadata in large open-source ecosystems such as Hugging Face. Consequently, CodeCarbon and CarbonTracker both remain limited for large-scale assessments without complete training metadata.

Taken together, existing studies show that AI training can generate substantial emissions, but the evidence remains fragmented and largely limited to individual models or voluntary disclosures. Such snapshots cannot reveal the aggregate footprint of the tens of thousands of models now hosted and shared globally, making it difficult to assess the scale of AI's environmental impact or design effective mitigation strategies. To address this gap, we use Hugging Face, the largest open repository of AI models, as a vantage point for estimating model-level training emissions at scale.

## 3. Estimating Training Emissions

Hugging Face hosts more than two million models, of which approximately 1.7 million are publicly accessible. Many entries are re-uploads, format conversions, or quantized variants that do not involve new training, while others lack essential training information. After filtering, we retained widely used models, resulting in 5,227 models with more than 5,000 downloads. Our primary analysis focuses on this >5,000 downloads group.

### 3.1. Ideal Runtime-Based Estimation Model

In an ideal scenario, if the computational power of the supercomputer used for training is known ($P_{\text{comp}}$), together with the total training time ($T_{\text{comp}}$) and the carbon intensity of electricity in the training region ($EF_{\text{region}}$, measured in kgCO$_2$/kWh), the training emissions can be estimated as

$$E_{\text{train}} = P_{\text{comp}} \times T_{\text{comp}} \times EF_{\text{region}}. \qquad (1)$$

However, few models disclose all the complete information,

and obtaining accurate data on the carbon footprint of supercomputing centers is even harder. Therefore, alternative strategies are required.

**Estimating Computational Power.** We approximate the effective computational power of the supercomputer through the following decomposition:

$$P_{\text{comp}} \approx N_{\text{GPU}} \times P_{\text{GPU}}^{\text{eff}} \times \text{PUE}, \qquad (2)$$

where $N_{\text{GPU}}$ denotes the number of GPUs employed during training, and $P_{\text{GPU}}^{\text{eff}}$ represents the effective average power draw per GPU (in kW). We define $P_{\text{GPU}}^{\text{eff}} = P_{\text{GPU}} \times R_{\text{eff}}$, where $P_{\text{GPU}}$ is the nominal or rated power consumption of the GPU (often approximated by its Thermal Design Power, TDP), and $R_{\text{eff}}$ is a runtime utilization factor that accounts for the gap between theoretical peak and actual workload efficiency. The term PUE stands for the Power Usage Effectiveness of the data center, which accounts for the additional overhead of cooling and infrastructure and typically ranges between 1.2 and 1.7 (Cae Lighting, 2025).

**Estimating Training Time.** The training time is estimated based on the overall computational workload required, expressed in floating-point operations (FLOPs). For a given model, the total training FLOPs is denoted by $F_{\text{train}}^{\text{total}}$. Assuming knowledge of GPU throughput, the base training time can be approximated as

$$T_{\text{base}} = \frac{F_{\text{train}}^{\text{total}}}{\theta_{\text{GPU}} \times N_{\text{GPU}} \times R_{\text{eff}}}, \qquad (3)$$

where $\theta_{\text{GPU}}$ is the sustained throughput per GPU in FLOPs per second (e.g., $3.12 \times 10^{14}$ FLOPs/s for NVIDIA A100 SXM under TF32 as shown in Table 6 of Appendix E.1), $N_{\text{GPU}}$ is the number of GPUs, $R_{\text{eff}}$ is the runtime utilization efficiency. Since training often involves restarts, debugging, and warm-up cycles, we also incorporate a **time amplification factor** $A_{\text{time}} \geq 1$, yielding $T_{\text{comp}} = T_{\text{base}} \times A_{\text{time}}$.

**Final Estimation Model.** Combining Eqs. (1), (2), and (3), the training-related carbon emissions of Hugging Face models can be estimated as

$$E_{\text{train}} \approx \underbrace{\left( N_{\text{GPU}} \times P_{\text{GPU}} \times R_{\text{eff}} \times \text{PUE} \right)}_{P_{\text{comp}}}$$

$$\times \underbrace{\left( \frac{F_{\text{train}}^{\text{total}}}{\theta_{\text{GPU}} \times N_{\text{GPU}} \times R_{\text{eff}}} \times A_{\text{time}} \right)}_{T_{\text{comp}}} \times EF_{\text{region}}$$

$$= \frac{P_{\text{GPU}}}{\theta_{\text{GPU}}} \times \text{PUE} \times F_{\text{train}}^{\text{total}} \times A_{\text{time}} \times EF_{\text{region}}. \qquad (4)$$

Eq. (4) represents our estimation framework for model-level training emissions. It is physically consistent and captures

the key drivers of training-related emissions: $\frac{P_{\text{GPU}}}{\theta_{\text{GPU}}}$ is effective energy per compute. $F_{\text{train}}^{\text{total}}$ reflects model size and training iterations. PUE represents data center overhead, accounting for cooling and distribution losses. $A_{\text{time}}$ as a runtime amplification factor captures parallelization inefficiencies, communication overhead, and system-level delays. $EF_{\text{region}}$ translates consumed energy into carbon emissions based on the local electricity mix. In short, Eq. (4) decomposes training emissions into *hardware × efficiency × computation × runtime amplification × environment*.

**FLOPs and actual energy usage are not always directly correlated.** In practice, the relationship between them is mediated by multiple factors. Some of these effects are implicitly captured in our framework through the efficiency term ($P_{\text{GPU}}/\theta_{\text{GPU}}$) and the runtime amplification factor $A_{\text{time}}$. However, extending such analysis to the full deployment stage would require additional factors beyond training compute, such as repeated experimentation, hyperparameter search, and ablation runs, which are often not systematically disclosed. Therefore, this work focuses on **the realized cost of released models** as a conservative but robust accounting boundary. Realized costs refer to the training cost attributable to a finalized, released model instance, based on repository-level information. This also aligns with Hugging Face's reporting practices, where emissions or other information are typically disclosed at the level of the finalized model repository.

**AI Training Carbon Intensity.** While the direct estimation of training emissions is informative, it may not always be intuitive for practitioners. Eq (5) provides a simplified framework for quantifying training emissions, and it can be further abstracted by grouping all factors except $F_{\text{train}}^{\text{total}}$ into a single coefficient. We define this coefficient as the *AI Training Carbon Intensity (ATCI)*, which represents the average carbon emission per compute:

$$\text{ATCI} \approx \frac{P_{\text{GPU}}}{\theta_{\text{GPU}}} \times \text{PUE} \times A_{\text{time}} \times EF_{\text{region}}. \quad (5)$$

Similar to the regional emission factor $EF_{\text{region}}$, which translates electricity into carbon emissions based on power grid composition, ATCI translates compute into carbon emissions by integrating hardware efficiency, data center overhead, runtime amplification, and regional carbon intensity. In other words, ATCI can be interpreted as "environment cost per compute" for model training.

We calculate the ATCI at the model level across a large collection of HF models. To further validate this index, we regress observed training emissions in FLOPs, emission factors, and hardware families (see Appendix E.2). The regression results provide empirical support for the conceptual decomposition of ATCI in Eq. (5). As shown in Figure 2, under a log–log specification, training emissions exhibit ap-

proximately linear scaling with both training compute and regional emission factors, as reflected by the coefficients of $\log(F_{\text{train}}^{\text{total}})$ (0.83) and $\log(EF_{\text{region}})$ (0.85). Significant hardware-specific effects (accelerator families) also highlight the role of accelerator efficiency in shaping ATCI. Hence, ATCI is a theoretical abstraction of the environment cost per compute. It can serve as an *important quantitative metric* for comparing the sustainability efficiency of training across models, providing the community with a practical reference even in the absence of complete disclosures.

**Carbon Intensity of Regional Grids among Models.** To estimate the carbon emissions associated with model training, we assign each model a regional electricity carbon intensity based on the best available geographic information: (i) When the model card specifies the training region or provides a specific emission factor, that value is used directly. (ii) In the absence of such disclosures, the region is inferred from the training organization's compute infrastructure or institutional affiliations. Regional factors follow the *Carbon Intensity of Electricity Generation* dataset from Our World in Data (Ember & Our World in Data, 2025). In cases where region information is also missing or indeterminate, we use the global average carbon intensity of $0.445$ tCO$_2$/MWh, consistent with IEA guidelines (IEA, 2024).

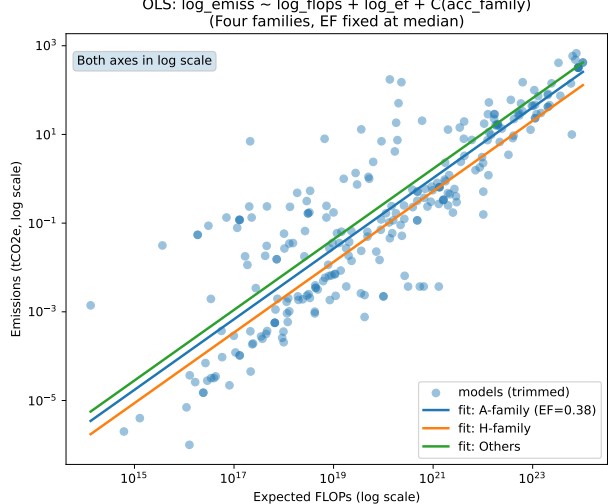

*Figure 2.* Scatter plot of estimated training emissions versus expected FLOPs, with regression fits for different accelerator families (A: NVIDIA A100/A800, H: H100/H800, Others). Both axes are log-scaled. The fitted model is $\log(E_{\text{train}}) = -39.25 + 0.85 \log(EF_{\text{region}}) + 0.83 \log(F_{\text{train}}^{\text{total}}) - 0.83\, I\{\text{H-family}\} + 0.63\, I\{\text{Others}\}$. Results indicate $\sim$0.83 FLOPs elasticity. Relative to the A-family (baseline), the H-family shows about 56% lower emissions. The "Others" exhibits roughly 88% higher emissions. A unified PUE and time amplification factor are assumed due to missing data center disclosures.

## 3.2. Training FLOPs Estimation

One of the critical quantities is the total training compute $F_{\text{train}}^{\text{total}}$, expressed in FLOPs. For transformer-based NLP models (e.g., BERT, GPT, LLaMA), we use the standard FLOPs approximation, FLOPs $\approx c \times N_{\text{params}} \times N_{\text{tokens}}$, where $c$ reflects the relative cost of attention and feed-forward operations (details in Appendix C). Empirical studies suggest $c$ typically falls in the range 5 to 8, and we adopt $c = 6$ as a conservative baseline, while sensitivity analyses with an extended range are reported in Appendix G. For CV and MM models, we apply architecture-specific heuristics (details in Appendix D). For Vision Transformers (ViTs) and CLIP models, FLOPs are estimated from patch embeddings and Transformer blocks, with training FLOPs approximated as six times the single-step inference cost; for CLIP, we apply a $1.1\times$ adjustment to account for the language branch. For diffusion models (e.g., Stable Diffusion, DiT), FLOPs are calculated by summing the convolution, self-attention, and cross-attention costs across denoising steps. For large multimodal Transformers that process image-text tokens with LLM-like backbones, we approximate compute as FLOPs $\approx 6 \times N_{\text{params}} \times N_{\text{tokens}}$, analogous to NLP models. Details of architecture-specific formulas, corrections for fine-tuning and Mixture-of-Experts (MoE) structures, and our imputation strategy for missing parameters are provided in Appendix C and D.

## 3.3. Handling Missing Values

**Three-Tier Strategies.** Emission estimation relies on partially disclosed information, which we cross-validate against multiple sources. We adopt a three–tier framework: 1) Tier 1 with rich disclosures (hardware type, GPU hours, or FLOPs). Emissions are computed from electricity use (GPU hours $\times$ power $\times$ grid factor) and from FLOPs–based inference, serving as calibration points (Appendix E.1); 2) Tier 2 with partial disclosures (e.g., FLOPs only). We impute missing values using representative hardware efficiencies and average overheads (Appendix E.2). Representative cases in Figure 2 also show how disclosure profiles map to estimation strategies and how regressions link FLOPs to emissions across hardware generations; 3) Tier 3 with minimal information (e.g., parameters only) or no usable training-related information. When parameter counts are available, emissions are approximated using parameter-based regressions (Appendix E.3); otherwise, we impute emissions using the average level of comparable models.

## 3.4. Uncertainty Propagation

Our estimation framework involves several quantities that carry measurement or imputation uncertainty. Since these variables enter multiplicatively in Eqs. (3)–(5), we propagate uncertainty (Coleman et al., 2009) using the standard first-

order relative-error formulation (Tyagi & Haan, 2001) for products:

$$\frac{\Delta E}{E} \approx \sqrt{\sum_i \left( \frac{\Delta x_i}{x_i} \right)^2}, \tag{6}$$

where $x_i \in \{F_{\text{train}}^{\text{total}}, P_{\text{GPU}}, \theta_{\text{GPU}}, A_{\text{time}}, \text{PUE}, EF_{\text{region}}\}$. The expression in Eq. (6) shows that the uncertainty in $E_{\text{train}}$ is governed by the combined relative errors of the multiplicative factors that define the training emissions. The resulting uncertainty structure is summarized in Appendix F.

Importantly, building a scalable and empirically grounded testbed for carbon accounting of AI models is inherently challenging but essential. While uncertainty is unavoidable due to incomplete disclosures, we aim to provide transparent, verifiable, and well-scoped estimates within a clearly defined accounting boundary.

## 4. Results

### 4.1. Training Emissions Estimation

Our reporting results follow standard significant-digit rules: aggregate emissions are given with at most two significant digits. Thus, our estimates indicate that, as of August 2025, training 5,227 models with more than 5,000 downloads has resulted in cumulative emissions of approximately $6 \times 10^4$ tCO$_2$e with an uncertainty of $\pm 2.4 \times 10^4$ tCO$_2$e, consistent with the propagated error in Eq. (6) (See details in Appendix F). We compare average ATCI and model-level emissions across modalities and training types in Table 1.

**CV & MM models exhibit higher training emission intensity than NLP.** CV's average ATCI is 0.16 tCO$_2$e/EFLOP versus NLP's 0.13 tCO$_2$e/EFLOP, indicating that per unit compute of vision training tends to translate into more energy and carbon emissions. This gap comes from heavier data pipelines and lower hardware efficiency in vision workloads (e.g., large image/video batches, augmentation, diffusion/decoder-only VAEs, and higher I/O/memory pressure that reduces accelerator utilization), as well as the prevalence of multi-stage training (pretrain + alignment + SFT) for Vision-Language Models (VLMs).

**Emission differences between foundation models (or individual models) and finetuned models.** The results highlight a clear divergence between Foundation & Individual models and Finetuned models in both emission intensity and their aggregate emissions. Finetuned models exhibit a higher mean ATCI (0.23 vs. 0.14 tCO$_2$e/EFLOPs), suggesting that each unit of computation in downstream training typically incurs greater carbon emissions. This pattern aligns with the typical deployment environments: large foundation and standalone models are often trained on

*Table 1.* Emission indicators and repository counts.

*(a)* Model-level $CO_2$e Emission Indicators

| Category | Mean ATCI (t/EFLOPs) | Mean (t) | Total ($10^4$ t) |
|---|---|---|---|
| Foundation & Individual | 0.14 | 12 | 5.6 |
| Finetuned models | 0.23 | 8 | 0.4 |
| CV & Multi-Modal (MM) | 0.16 | 12 | 2.3 |
| NLP | 0.13 | 11 | 3.6 |

*(b)* Repository Counts by Tier (Downloads>5,000)

| Tier | NLP Repos | CV/MM Repos |
|---|---|---|
| Tier 1 | 390 | 352 |
| Tier 2 | 944 | 1679 |
| Tier 3 | 3053 | 220 |

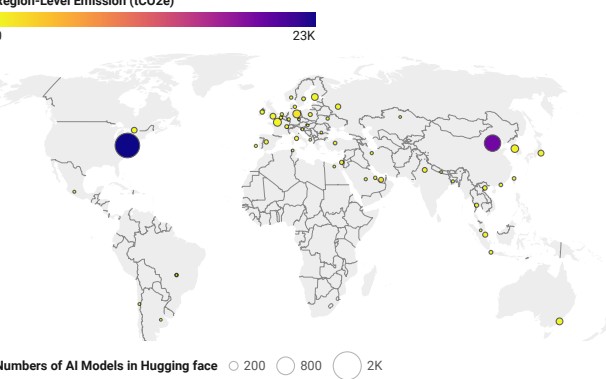

*Figure 3.* Global cumulative training emissions of AI models (downloads>5,000).

centralized, energy-efficient clusters with optimized hardware utilization and cleaner grid mixes, whereas finetuning workloads are more widely distributed across smaller-scale, less efficient, and often metadata-poor computing environments, which inflates per-EFLOP carbon intensity. Despite their higher ATCI, finetuned models contribute only a minor share of the absolute value of total emissions, as the computational scale of foundation-model pre-training overwhelmingly dominates. Overall, while finetuning tends to be "higher emissions per EFLOP," the majority of AI's training-related emissions is still driven by a relatively small number of extremely compute-intensive foundation-model runs.

**ATCI.** We further interpret the significance of ATCI, the ratio of training emissions per compute. ATCI captures the carbon efficiency of model training pipelines by abstracting away from model size or absolute compute cost, thereby providing a normalized metric for comparing across modalities and training paradigms. Overall, the results highlight that (a) *modality matters*: vision/multimodal training is more carbon-intensive per compute; (b) *lifecycle practice matters*: finetuned variants exhibit higher per-checkpoint emissions not only because they undergo repeated downstream training and alignment cycles, but also because they typically run on less energy-efficient hardware environments; (c) *hardware matters*: according to the empirical results in Figure 2, models with the H-family accelerators show about $56\%$ lower emissions compared to those with A-family accelerators after controlling for compute and electricity emission factors. Model-level ATCI can serve as a standard reporting metric for comparing the intensity of the carbon footprint and the training efficiency of AI models and products.

### 4.2. Training Emissions across Regions and Time

**Region.** As shown in Figure 3, regional aggregation reveals an uneven distribution of training emissions. The United States dominates the landscape ($2.3 \times 10^4$ tCO$_2$e), followed by China ($2.0 \times 10^4$ tCO$_2$e). In contrast, most European countries (e.g., the United Kingdom, France, Italy, Finland), as well as Canada and Australia, host repositories but generate small emissions per model, indicating lighter-weight workloads or lower-compute research practices.

**Temporal Evolution.** In Figure 4, training emissions of models (downloads 5000+) on Hugging Face have esca-

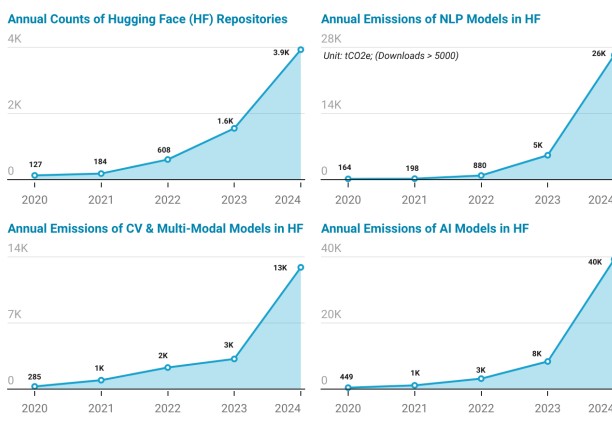

*Figure 4.* Annual training emissions (unit: ton $CO_2e$) of AI models (downloads>5,000) in HF from 2020 to 2024

lated sharply over time. From 2020–2021 to 2024–2025, annual emissions increased from only $\sim 4.5 \times 10^2$ tCO$_2$e to more than $4.0 \times 10^4$ tCO$_2$e, reflecting nearly two orders of magnitude growth within five years. The composition of these emissions also shifted substantially. Early periods were dominated by CV and MM models, but NLP activity expanded rapidly between 2022 and 2024, becoming the largest contributor during this interval. In the most recent period (2024–2025), CV and MM models once again surpassed NLP models due to a surge in vision and multimodal releases. Together, these trends reveal both the accelerating pace of model training and the evolving distribution of computational demand across major research domains.

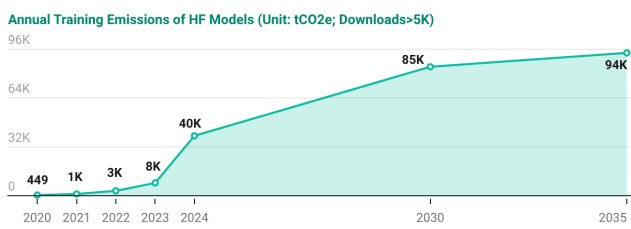

*Figure 5.* Projected training emissions of HF models with over 5,000 downloads from 2024 to 2035.

**Projected Emissions.** According to the IEA projection (IEA, 2024), the electricity demand of global AI data centers is expected to rise from about $1.3\%$ of global electricity demand in 2024 to nearly $2.8\%$ by 2030 (and stabilizing nearly $3.1\%$ by 2035). Figure 5 similarly illustrates a projected increase in model emissions. Our estimates show that models with over 5,000 downloads will grow from $\sim 4.0 \times 10^4$ tCO$_2$e in 2024 to $\sim 9.4 \times 10^4$ tCO$_2$e in 2035.

### 4.3. Carbon Disclosure Quality of AI Models

Among the more than two million repositories on Hugging Face, only 2,422 include a `co2_eq_emissions` field, and fewer than 200 provide additional energy or emissions details in their `README`. In total, less than $0.2\%$ of models disclose any environmental footprint, underscoring a substantial transparency gap (see Figure 6 of Appendix A). Even within the disclosed set, many suffer from inconsistent multi-source reporting and erroneous values, limiting their reliability. Table 2 highlights several representative model cases. It compares disclosed values from technical reports or Hugging Face metadata with our estimates, showing that our results are generally consistent with disclosures.

Due to the lack of detailed disclosures for most models, we approximate some missing quantities using industry or region-level averages for tier 2 and tier 3 models, which inevitably introduces uncertainty. Therefore, we conduct uncertainty decomposition and pseudo-missingness experiments to show the feasibility of our approach in Appendix G and H. Results indicate that these estimation errors remain within a reasonable range: Tier 2 estimates are highly stable (90% of estimates differ by no more than $\sim 1.2\times$), while Tier 3 remains informative despite minimal metadata (90% of estimates within $\sim 2\times$).

**Error on Models with Disclosed Emissions.** To evaluate the accuracy of our framework against ground-truth disclosures, we analyze 292 models that publicly report their total training emissions (see Appendix A and G). To ensure robustness, we exclude unreliable disclosures and numerically unstable cases, and adopt a robust trimming procedure to mitigate the impact of heavy-tailed outliers. Relative errors are defined as $\mathrm{RE}_i = |\hat{E}_i - E_i|/E_i$. To obtain a stable evaluation, we perform symmetric trimming, retaining the

*Table 2.* Illustrative comparison between disclosed and our estimated training emissions (units: tCO$_2$e). Here, Llama 2 series include meta-llama/Llama-2-13b-hf, meta-llama/Llama-2-70b-hf, and meta-llama/Llama-2-7b-hf; CodeLlama series covers the 7B, 13B, and 34B base, Python, and Instruct variants.

| Model series | | | |
|---|---|---|---|
| Model series | Estimation | Disclosed | Source |
| Llama 2 | 412 | 384 | (Touvron et al., 2023) |
| CodeLlama | 72 | 65 | HF disclosed |

| Single model | | | |
|---|---|---|---|
| Model | Estimation | Disclosed | Source |
| Meta Llama 2 (7B) | 33 | 31 | (Touvron et al., 2023) |
| Meta Llama 2 (13B) | 52 | 62 | (Touvron et al., 2023) |
| Meta Llama 2 (70B) | 327 | 291 | (Touvron et al., 2023) |
| Meta-Llama 3 (70B) | 1,010 | 1,900 | HF disclosed |
| Meta Llama 3.1 405B | 8,176 | 8,930 | (Maslej et al., 2025) |
| Bloom | 25 | 24.7 | (Luccioni et al., 2023) |
| OLMoE-1B-7B-0924 | 30 | 18 | (Morrison et al., 2025) |
| stable-diffusion-v1 | 13 | 11.25 | HF disclosed |
| sam-vit-base | 3 | 2.8 | HF disclosed |
| sam2-hiera-small | 5 | 3.89 | HF disclosed |
| bioclip | 0.20 | 0.13 | HF disclosed |
| stable-diffusion-2 | 17 | 15 | HF disclosed |
| stable-video-diffusion-img2vid | 13 | 19 | HF disclosed |
| stable-diffusion-v1-5 | 13 | 11.25 | HF disclosed |

central 95% of samples by excluding the lowest and highest 5% of relative-error values. The evaluation yields the results in Table 3, which indicate that the majority of models exhibit stable and accurate emission estimates, with approximately 74% and 82% of models falling within $\times 2$ and $\times 3$ of their disclosed values, respectively.

*Table 3.* Robust evaluation on models with disclosed emissions.

| Metric | Value |
|---|---|
| MAPE | 0.42 |
| Median RE | 0.32 |
| Hit rate ($\times 2$ / $\times 3$) | 0.74 / 0.82 |

## 5. Conclusions

This paper presents a FLOPs-based framework to estimate training-related carbon emissions of Hugging Face models at scale. Our analysis shows that even within the open-source ecosystem, cumulative training emissions already reach the order of $10^4$–$10^5$ tons of CO$_2$e, comparable to the annual carbon footprint of thousands of EU residents or the annual emissions of tens of thousands of passenger cars. This highlights both the urgency of standardized disclosure and the value of open repositories as anchors for industry-scale carbon accounting.

**Limitation and Future Work.** Our study presents the systematic accounting of training-related carbon emissions for mainstream models hosted on Hugging Face. These results provide a useful reference point for researchers, practitioners, and the public in understanding the environmental costs of AI. At the same time, several important limitations remain, highlighting directions for future work. First, our analysis focuses exclusively on open-source models.

A large fraction of the most influential models are proprietary, and their training processes and energy consumption remain undisclosed. Existing reports suggest that these closed-source models may contribute substantially to overall emissions, likely exceeding the footprint of the open-source community. Second, we focus only on training emissions. Yet training is only one part of the picture. Research activities that do not yield a final deployed model also consume considerable resources, and inference at deployment scale is expected to dominate AI's long-term energy demand. Understanding the emissions from inference workloads will require complementary approaches, such as analyzing data center expansion, hardware deployment statistics, and the size of the inference services market. Third, our study does not attempt to capture the full lifecycle emissions of AI systems. A complete assessment would account for the embodied carbon from hardware manufacturing, research and experimentation, model training, and deployment-scale inference, as well as the accounting and attribution of such emissions across stakeholders. Developing standardized methodologies for lifecycle carbon accounting in AI remains an open and urgent challenge.

**Extension to inference emission estimations.** While our main analysis focuses on training, the framework can be extended to inference. The inputs can switch to inference-specific quantities: the power and throughput of the inference hardware (often different from training GPUs), the efficiency and batching characteristics of inference workloads, and the compute required per generated token. Once collecting these inputs, our framework can yield inference-emission estimates and inference emission intensity in exactly the same way as for training.

Our work is an initial step toward scalable estimation of emissions in the AI industry. By quantifying the training emissions of a large body of open-source models, we provide an empirical anchor that future studies can extend toward closed-source models, inference workloads, and full lifecycle assessments. Such progress is essential for aligning AI development with sustainability goals and for informing future AI regulation and policy.

## Acknowledgements

This research was partially funded by the Ministry of Education and Science of Bulgaria (support for INSAIT, part of the Bulgarian National Roadmap for Research Infrastructure).

## Impact Statement

A fundamental question is the scale of emissions generated by the AI industry. Until now, the community has lacked any empirical corpus large enough to enable systematic quantification. This paper provides the community with a new empirical foundation and enables, for the first time, an aggregated view of emissions across thousands of open models. On top of this empirical foundation, this paper also introduces a unified estimation pipeline designed to operate across heterogeneous metadata, inconsistent logging, and incomplete disclosures. This is important for making large-scale carbon accounting possible and goes beyond routine engineering. It provides the analytical framework required to make previously unquantifiable parts of the platform measurable. This is precisely the level needed to resolve the scale question the community currently cannot answer: whether the open-model ecosystem emits thousands, tens of thousands, or hundreds of thousands of tons of $CO_2e$. Our results provide the first concrete answer at the correct scale.

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

# Appendix

## A. Self-Disclosed Emissions in Hugging Face Platform

**Metadata Disclosure Landscape**    We evaluate metadata disclosure across all 5,227 models in our dataset. To construct this metadata repository, we systematically collected training-related information from three classes of sources:

1. **Official Hugging Face model cards**, including structured fields (e.g., `hardware`, `carbon_emissions`), author-provided notes, and embedded configuration snippets.

2. **Repository configuration files**, such as `config.json`, tokenizer/vision encoder configs, and architecture descriptors. These files provide parameter counts, layer depths, hidden sizes, patch sizes, and other FLOPs-relevant attributes.

3. **External authoritative sources**, including official technical reports, GitHub repositories, and arXiv papers referenced in the model cards. When multiple sources were available, we applied a deterministic priority order (direct disclosure → config-derived → paper-derived → regression-estimated).

*Table 4.* Metadata disclosure sparsity across the Hugging Face models (5,000+ downloads).

| Metadata Field | Count | Disclosure Rate |
|---|---|---|
| Training emissions (tCO$_2$e) | 292 | 5.58% |
| Electricity use (MWh) | 15 | 0.29% |
| Grid emission factor | 5 | 0.10% |
| Training region | 54 | 1.03% |
| GPU type | 955 | 18.25% |
| TPU Pod | 159 | 3.04% |
| Training runtime hours | 179 | 3.42% |
| Training device count | 414 | 7.91% |

Only 5–6% of models (with 5,000+ downloads) disclose energy or emission-related metadata. This structural sparsity is the primary source of uncertainty in open-source carbon accounting. For all models with any disclosed training information

(FLOPs, electricity use, grid factors, or total emissions), we have compiled a detailed comparison table containing disclosed quantities and our reconstructed estimates.

**Summary of Models with Self-Disclosed Emissions**   The set of 292 models that self-disclosed their training emissions includes series such as *Bloom*, *CodeLlama*; recent Meta Llama 3/3.1/3.2 and Llama 4 variants (e.g., *meta-llama/Llama-3.1-405B*, *meta-llama/Llama-3.1-70B*, *Meta-Llama-3-70B*); AllenAI's *OLMo* and *OLMoE* models (e.g., *allenai/OLMo-7B-hf*, *allenai/OLMo-2-1124-13B-Instruct*); EleutherAI's *GPT-NeoX-20B*; image and video models from Stability AI (e.g., *stable-diffusion-2*, *stable-video-diffusion-img2vid* and related variants); and smaller models such as *ModernBERT* variants, rerankers, and tiny classifiers.

| | Counts |
|---|---|
| All HF Repositories | 2,099,013 |
| Carbon Emission Disclosed in HF Carbon Emission Modules | 2,422 |
| Carbon Emission Disclosed in Readmes | 126 |

*Figure 6.* **Total Amount of Models with Self-Disclosed Emission in Hugging Face.** Out of more than 2.1 million repositories, only 2,422 include a structured carbon emissions field (`co2_eq_emissions`) and just 126 mention energy use or emissions in their README files, highlighting a disclosure rate below 0.2%.

**Curating Models with Reliable Disclosed Emissions:**

- **Total number of repositories in the accounting scope**: In Hugging Face, there are 6,638 repositories with downloads $\geq$ 5K as of August 2025, forming the initial candidate pool for our analysis.

- **Deduplication**: After deduplication (removing mirrors/duplicates), we retain 5,227 repositories (and corresponding models), meaning 1,411 repositories are removed.

- **With `co2_eq_emissions` field or direct emission disclosures**: This leaves 338 models with directly disclosed emissions information, which means 4,889 models are removed.

- **Quality control**: For data quality definitions, unreliable disclosures indicate clear inconsistencies (e.g., unit errors, orders-of-magnitude errors, or typos). Numerically unstable means values that yield implausible results under cross-checks (e.g., unrealistically low). After manual verification, we retain 292 models, which means we removed 46 models in total (13 models due to numerically unstable cases, and 33 models due to unreliable cases)

Here we provide several concrete examples of cases of numerically unstable:

- **Unrealistically high**: One example is `nvidia/Cosmos-Reason1-7B`, for which a disclosure reported 5,380 metric ton emissions with metadata disclosing $3.26 \times 10^{21}$ training FLOPs associated with training. Given its 7B scale and training flops, this implies an energy intensity far above comparable models under similar hardware assumptions. It is nearly ten times the estimated emissions of GPT-3 (175B Parameters, 552.1 metric tons emissions, training flops: $3.14 \times 10^{23}$). We therefore excluded it from the disclosed value, considering it numerically unstable.

  Importantly, we found that this value is no longer present in the current repository now, suggesting that the original disclosure has been revised or removed. The earlier version can still be accessed via the AWS Marketplace listing and the Hugging Face mirror repo (`unsloth/Cosmos-Reason1-7B`).

- **Unrealistically low**: The `co2_eq_emissions` field of `KoalaAI/Text-Moderation` shows that it only emitted 0.04 g$CO_2$ eq emissions during training. Given that the model is based on a DeBERTa-scale architecture, the disclosed value is only a few seconds of GPU execution when converted to energy consumption. We can perform a simple back-of-the-envelope check: Assuming a typical carbon intensity of 0.4 t$CO_2$/MWh, 0.04 g $CO_2$ corresponds to only 0.0001 kWh of energy. This is equivalent to nearly 1 second of GPU runtime under typical accelerator power (e.g., A100 at 400W), which is inconsistent with any full training or fine-tuning process. As a result, we removed such unrealistically low value to avoid systematic underestimation.

## B. Data Collection and Agent Processing Pipeline

**Automated Crawling.** We collect heterogeneous metadata from Hugging Face model repositories and associated documentation. The crawler reads repository descriptors (`README.md`, model cards, metadata CSVs, configs.json) and extracts candidate fields including *hardware type*, *GPU/TPU counts*, *training duration*, and especially *training FLOPs*. For FLOPs disclosures, we implemented robust parsing functions that can handle varied numeric expressions (e.g., shorthand "2k", "1.2M", or scientific notation such as "$5 \times 10^{21}$"), ensuring standardized floating-point values for downstream estimation. All extracted fields are normalized and stored in structured CSV/JSON tables, providing a consistent basis for regression analysis and emission estimation.

**Repository Deduplication.** To avoid double-counting emissions from mirrored repositories, we applied a systematic deduplication rule: when both an official repository and an `unsloth/...` mirror exist, the mirror is dropped unless the discrepancy in reported values is negligible ($\leq 0.1\%$), in which case the `unsloth` version is retained as canonical. In addition, we excluded derivative artifacts such as GGUF or quantized models (e.g., 4bit/8bit, AWQ, GPTQ/PTQ/NF4/FP8/Q4/Q5) since they represent deployment optimizations rather than independent training runs. These filters ensure that only unique, training model entries are preserved in the dataset.

**Agent Workflow.** To handle inconsistent disclosures and missing fields, we developed an LLM-based agent workflow (GPT-4o) that performs: (i) **hardware recognition**, mapping noisy or aliased strings to canonical GPU/TPU families; (ii) **unit normalization**, distinguishing between wall-clock hours and GPU-hours using contextual cues; (iii) **cross-file integration,** employing a dedicated **web search agent** to locate and retrieve corresponding technical reports or project website released by model developers, which were then cross-validated against Hugging Face metadata and incorporated into the final dataset. We merge all findings with regional emission factor datasets. Ambiguous cases (e.g., extreme FLOPs values, unclear unit conventions) were flagged for manual inspection by human annotators.

**Human Verification.** To ensure reliability, five independent human annotators reviewed a stratified subsample of repositories. They checked accelerator mappings, parsed FLOPs statements, and validated whether durations corresponded to GPU-hours or wall-clock hours. Annotators resolved edge cases such as conflicting information across README text and metadata tables. Inter-annotator agreement was calculated to calibrate the agent's confidence thresholds.

**Data Integration.** All sources (GPU/TPU metadata, FLOPs estimates, and regional emission factors) were merged into unified tables via normalized identifiers. Duplicate columns and conflicting values were harmonized, and each record carries diagnostic notes (e.g., method of estimation, source of FLOPs, reasons for imputation). This enables transparent traceability of every emission estimate. The final dataset consists of harmonized records with accelerator type, count, training duration (direct or imputed), FLOPs used, power draw, regional EF, and estimated emissions ($tCO_2e$). All records include provenance notes indicating whether values were obtained via direct disclosure, agent inference, or human annotation.

## C. NLP Models Training FLOPs Estimations

OpenAI's scaling law study (Kaplan et al., 2020) introduced the widely used approximation for training compute of large-scale language models:

$$\text{FLOPs} \approx c \times N_{\text{params}} \times N_{\text{tokens}}, \tag{1}$$

where $N_{\text{params}}$ is the number of model parameters, $N_{\text{tokens}}$ the number of training tokens, and $c$ a constant reflecting the balance between attention and feed-forward operations. We refine this approximation to account for different categories:

- **Architecture type.** Encoder-only models (e.g., BERT), decoder-only models (e.g., GPT, LLaMA), and encoder–decoder models (e.g., T5, BART) differ in the ratio of feed-forward to attention compute, which shifts $c$ within the baseline range of 5–8.

- **Parameter-efficient fine-tuning (PEFT).** For methods such as adapters and LoRA, only a fraction of parameters are trainable. We therefore rescale the effective parameter count to reflect $N_{\text{trainable}}$, while partially accounting for frozen weights that still incur forward-pass compute during backpropagation.

- **Mixture-of-Experts (MoE).** For MoE architectures, dense parameter count does not represent the actual compute cost. We instead replace $N_{\text{params}}$ with the number of *active* parameters per token, determined by the top-$k$ experts selected during routing, and introduce a routing overhead correction.

Unless specified, we adopt $c = 6$ as a conservative baseline for the main analysis, while sensitivity analyses over the full range are reported in this supplement. In practice, we estimate training compute for transformer-based NLP models by combining structural information with training configuration metadata extracted from Hugging Face model cards, repository documentation, and associated papers. This process is automated in our analysis pipeline and implemented in several steps:

**1) Model classification and parameter extraction.** Each model is classified as encoder-only (e.g., BERT), decoder-only (e.g., GPT, LLaMA), or encoder–decoder (e.g., T5). When available, we directly record the number of trainable parameters ($N_{\text{params}}$). If parameters are missing, we infer them from architecture descriptors such as hidden size, number of layers, and attention heads.

**2) Effective parameter count adjustments.** For pretraining we set $N_{\text{params}}$ to the full parameter count. For others, we distinguish:

- **Full-parameter Fine Tuning (FT)**: $N_{\text{params}}$ is the full count.

- **Parameter-efficient Fine Tuning (PEFT)** (e.g., LoRA/adapters): we substitute $N_{\text{params}}$ by the number of *active trainable* parameters $N_{\text{trainable}}$ and include a forward-pass reuse factor since frozen weights still incur inference-side compute during backprop. Concretely,

$$\text{FLOPs}_{\text{base,PEFT}} \approx c_{\text{arch}}\big(\alpha_{\text{frozen}} N_{\text{frozen}} + N_{\text{trainable}}\big) \times N_{\text{tokens}}, \tag{2}$$

  with $\alpha_{\text{frozen}} \in [0.2, 0.5]$ reflecting the proportion of frozen-path compute amortized in backward (empirical, task- and stack-dependent).

- **Mixture-of-Experts models**: we substitute the full parameter count with the number of active parameters per token, i.e., the sum of dense parameters and the top-$k$ experts activated per forward pass. Here, we replace $N_{\text{params}}$ by the *active* parameters per token, i.e.,

$$N_{\text{params}}^{\text{MoE}} \approx N_{\text{dense}} + \underbrace{k \cdot \frac{N_{\text{experts}}}{E} N_{\text{expert}}}_{\text{top-}k \text{ experts per token}}, \tag{3}$$

  where $k$ is the top-$k$ routing, $E$ is the number of experts per layer, and $N_{\text{expert}}$ the per-expert parameters.

**MoE adjustment.** MoE models introduce non-negligible system-level overheads beyond FLOPs. The calculation should capture the combined effect beyond memory and communication overhead. For example, All-to-All communication accounts for 34.1% of step time, up to 74.9% within a single MoE layer (Li et al., 2023). Communication accounts for 32% of total training time and 43.6% of forward time, with MFU dropping from 32.5% to 27.9% (Jin et al., 2026). Communication and activation memory are dominant bottlenecks, with utilization that can drop to less than 10% of peak FLOPs in some cases (Yuan et al., 2025). Meta empirical measurement (Appendix D): 160 vs 115 TFLOPs/GPU (dense vs MoE) on A100, corresponding to nearly 1.39× runtime/energy amplification (Artetxe et al., 2022). Based on the broader evidence above, these findings suggest that we should introduce a realistic correction factor $\alpha_{\text{route}}$ is 1.4×-2.0×, depending on system configuration. **Accordingly, we update our experiments by applying a general 1.5× correction factor to MoE models to account for their additional routing and system-level overheads.**

**3) Token accounting.** When $N_{\text{tokens}}$ is not directly reported, we infer it from dataset size and epochs, or reconstruct it from step geometry:

$$N_{\text{tokens}} \approx S \times G, \tag{4}$$

$$\text{where} \quad G = W \times A \times L \times B. \tag{5}$$

$S$ denotes the total number of training steps, $W$ denotes the world size (number of devices), $A$ denotes the gradient accumulation steps, $L$ denotes the average sequence length, and $B$ denotes the per-device batch size.

**4) Baseline FLOPs estimate.** Let $N_{\text{params}}$ denote the number of (active) trainable parameters and $N_{\text{tokens}}$ the number of training tokens effectively processed. The baseline lower-bound follows (Kaplan et al., 2020):

$$\text{FLOPs}_{\text{base}} \approx c_{\text{arch}} \times N_{\text{params}} \times N_{\text{tokens}}, \tag{6}$$

where $c_{\text{arch}} \in [5, 12]$ accounts for architectural differences in the ratio of attention and feed-forward compute. Encoder-only and decoder-only models default to 6, while encoder–decoder models use 7, with flexibility for further adjustments.

## D. CV and MM Models Training FLOPs Estimations

For multimodal models, we employ an architecture-specific methodology to estimate training FLOPs. Our automated analysis pipeline categorizes models into several primary architectures, including Vision Transformers (ViT), Contrastive Language-Image Pre-Training (CLIP) models, Convolutional Neural Networks (CNNs), Diffusion models, and Transformers. The core of this approach is extracting key architectural parameters from HuggingFace model cards and configuration files. For CNNs, however, we directly run the model with a randomized input tensor of a unified resolution to precisely calculate the single-step inference FLOPs.

Notably, this analysis excludes the computational cost of parameter-efficient fine-tuning (PEFT) techniques, such as LoRA and other adapters. While increasingly prevalent for model customization, the compute required for training these modules is typically several orders of magnitude smaller than that of full model pre-training or fine-tuning, rendering its contribution negligible in our large-scale carbon footprint assessment.

With $E$ as the number of training epochs and $I$ as the number of training images per epoch, we apply the following tailored estimation strategies for different architectures:

**1) ViT and CLIP models.** For Vision Transformer (ViT) based models, we first calculate the FLOPs for a single forward step by summing the contributions from the patch embedding layer and the subsequent Transformer blocks. Let $H, W, P, C$ be the input image height, width, patch size, and channels, respectively, and let $d, L, r$ be the model's hidden dimension, number of layers, and MLP expansion ratio. The number of input tokens is $N = \frac{H \cdot W}{P^2} + 1$ (including the [CLS] token).

The total MACs (Multiply-Accumulate operations) for one forward pass can be broken down as:

- **Patch Embedding**: $M_{embed} = H \cdot W \cdot C \cdot d$

- **Transformer Block**: The computation is dominated by the multi-head self-attention (MHSA) and the MLP layers, where $M_{MHSA} = 4Nd^2 + 2N^2d$ and $M_{MLP} = 2rNd^2$.

Thus, the total MACs for one single step of a ViT model can be expressed as:

$$M_{ViT} = M_{embed} + L \cdot (M_{MHSA} + M_{MLP}) = HWCd + L[(4 + 2r)Nd^2 + 2N^2d] \tag{7}$$

Based on the common heuristic that training FLOPs are approximately six times the inference MACs (accounting for a $3\times$ factor for the training procedure and a $2\times$ factor for converting MACs to FLOPs), the final FLOPs are:

$$F_{ViT} = 6 \times E \cdot I \cdot M_{ViT} \tag{8}$$

For CLIP models, we approximate the computational cost of the language branch as $10\%$ of the vision branch. Therefore, we apply a $1.1\times$ factor to the ViT result:

$$F_{CLIP} = 1.1 \times F_{ViT} \tag{9}$$

**2) Diffusion models.** For U-Net-based models (e.g., Stable Diffusion), the MACs for a single denoising step are calculated by summing the compute across all layers in the U-Net's down-sampling, middle, and up-sampling blocks. This includes contributions from 2D convolutions ($M_{conv}$), self-attention ($M_{SA}$), and cross-attention ($M_{CA}$) layers. The total FLOPs are then estimated as:

$$F_{Diffusion} = 6 \times E \cdot I \cdot (M_{conv} + M_{SA} + MCA) \tag{10}$$

For Diffusion Transformer (DiT) models, the calculation is analogous to that of ViT. The total FLOPs for a single step can be estimated by the sum of the patch embedding, the stack of $L$ Transformer blocks. The core computation within each DiT block, which includes self-attention, optional cross-attention, and an MLP, follows the same principles as the ViT block calculation.

$$F_{DiT} = 6 \times E \cdot I \cdot M_{DiT} = 6 \times E \cdot I \cdot [M_{embed} + L \cdot (M_{MHSA} + M_{MLP})] \tag{11}$$

**3) Transformers.** For Transformer-based models such as large vision-language models, where the architecture is predominantly a large language model processing multimodal tokens, the total training FLOPs are approximated as:

$$F_{Transformers} = 6 \times N \cdot D \tag{12}$$

where N represents the number of model parameters and D is the total number of tokens in the training data.

**Data Imputation Strategy.** Our automated pipeline may encounter models with incomplete configurations that lack the parameters necessary for FLOPs estimation. In such cases, we implement a prototype-based imputation strategy. Specifically, we pre-select a canonical or widely-recognized "prototype model" for each major architectural category (e.g., google/vit-base-patch16-224-in21k for ViTs). When a model is found to have missing parameters, the pipeline populates the missing fields with the corresponding values from the prototype model. For models where FLOPs cannot be calculated at all (e.g., due to a missing configuration file), we impute the final FLOPs value using the mean of all other models in the same category. This approach ensures the robustness and comprehensive coverage of our estimation process.

## E. Emission Estimation with Missing Values

To accommodate heterogeneous levels of disclosure across model repositories, we adopt a three–tier framework for training emission estimation:

- **Tier 1: Rich disclosures.** Models provide sufficient information required in Appendix. C and Appendix.C that is either directly disclosed or can be directly computed, such as *hardware type* (GPU/TPU family), reported *training GPU hours*, and/or total training FLOPs. In these cases, training duration and energy use can be established with the highest accuracy, enabling reliable emission estimation.

- **Tier 2: Partial disclosures.** Models have reported information for estimating the total FLOPs used in training, without hardware details or runtime information. Here, we estimate training emissions by assuming representative hardware efficiency values and average system overhead factors, mapping FLOPs into energy consumption under a standardized configuration.(see Appendix E.2)

- **Tier 3: Minimal disclosures.** Models report only the parameter count, with no FLOPs or hardware details available. For these cases, we rely on a parameter–based regression (see Appendix E.3) as a fallback, using cross–sectional elasticity estimates to approximate emissions from model scale. If no usable training-related metadata, including parameter counts, is available, we impute emissions using the average emission level of comparable models.

### E.1. Emission calculation for tier 1 models

We implement a unified estimator that integrates accelerator recognition, multi-node topology, overhead factors, and regional emission intensities to approximate training-related carbon emissions. The pipeline is designed to handle heterogeneous disclosures across model repositories, including cases with incomplete or ambiguous hardware information.

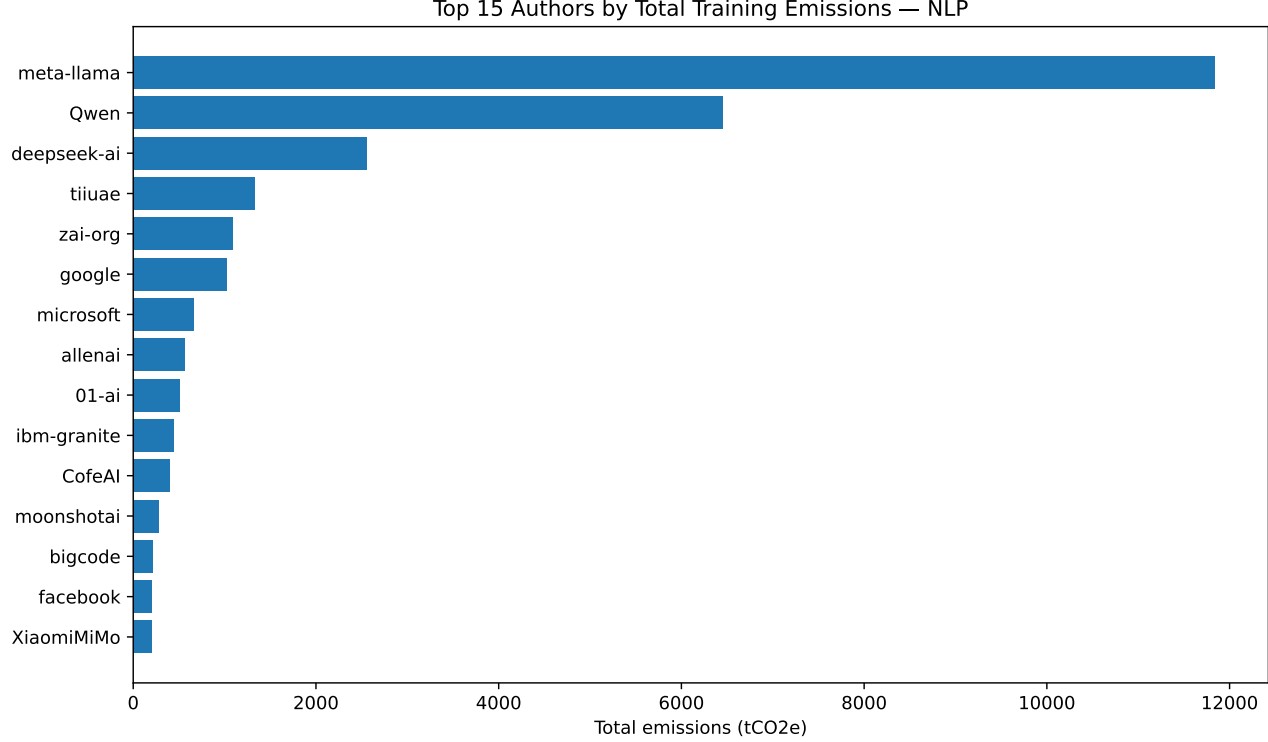

*Figure 7.* Top 15 Authors with Highest Estimated Training Emissions of Hugging Face NLP Models (5,000+ Downloads)

*Table 5.* Three–tier framework for handling missing values in training emission estimation.

| Tier | Available Information | Estimation Method & Accuracy |
|---|---|---|
| **1** | Hardware type, GPU hours, or total FLOPs (606 models) | Direct electricity use (GPU hours $\times$ power $\times$ grid factor) and FLOPs–based inference; *High* (calibration set) |
| **2** | FLOPs available but no hardware/runtime details | FLOPs mapped to energy using representative hardware efficiency and overheads; *Medium* |
| **3** | Parameter counts only | Parameter–based regression to approximate FLOPs and emissions; *Low* |

**Hardware Normalization and Accelerator Imputation Procedure** In the implementation, accelerators are mapped to a small set of *canonical families* with associated peak TFLOPS, average power, and efficiency: NVIDIA A100 / A100-80GB / A100-64GB / A800, H100 / H200 / H800, V100, A40, A30, T4, L4, RTX 6000 ADA, AMD MI250X / MI300X, and Google TPU V2 / V3 / V4 / V5E / V5P. Assignment proceeds as follows.

For Tier 1 (disclosed hardware), when model cards report training hardware type, these strings are used directly. If traininghardwaretype indicates TPU, the pod name (e.g., "v4-128", "v3-8") is parsed and mapped to a canonical TPU family; if the generation cannot be resolved, TPU V3 is used as a mid-range default. Otherwise, the device is treated as a GPU and is normalized using regex rules, matching patterns; the matched family is then used to look up peak TFLOPS, average power, and efficiency.

For Tier 2 and Tier 3 (imputed hardware), when metadata is incomplete, models with TPU hardware but ambiguous pod strings are assigned TPU V3 as a conservative default, and models known to use GPUs but lacking a resolvable training gpu type fall back to an A100-class assumption (A100 peak TFLOPS, ~0.30–0.35 efficiency, 400 W power) as a representative datacenter GPU.

AMD and TPU jobs are therefore not collapsed into NVIDIA families: MI250X and MI300X have their own TFLOPS/power

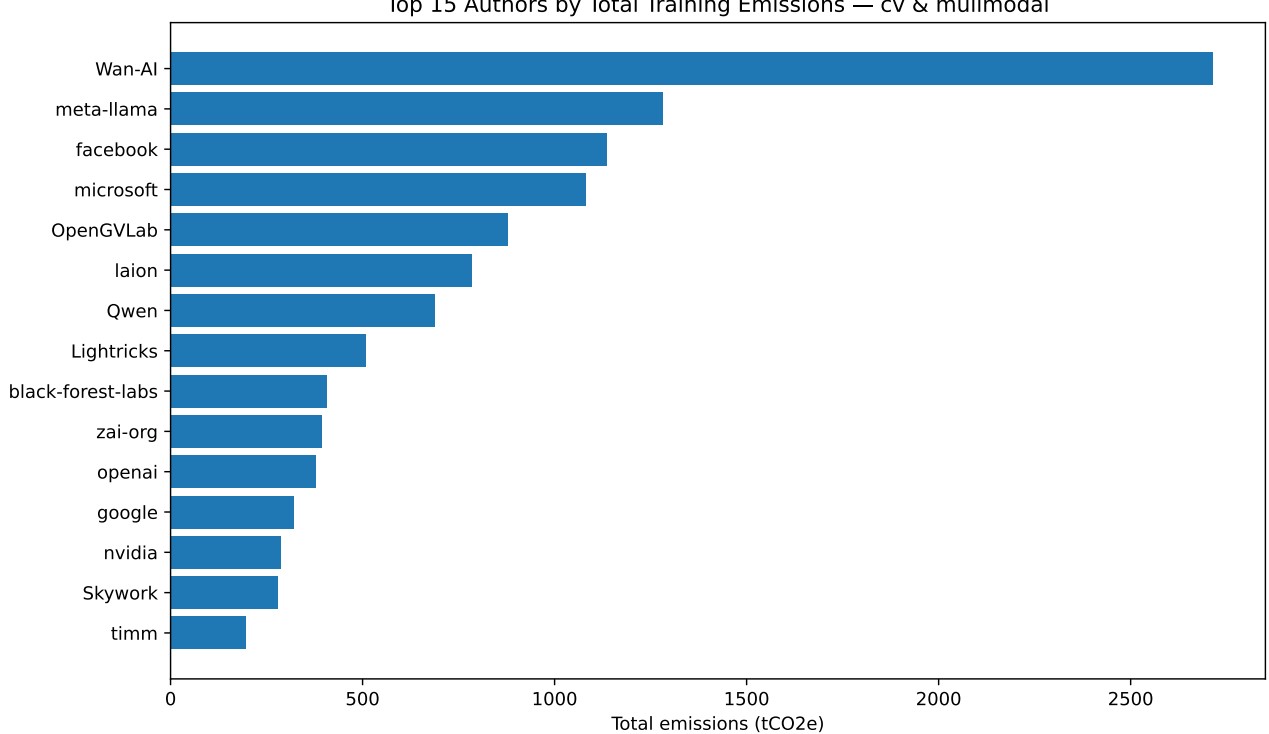

*Figure 8.* Top 15 Authors with Highest Estimated Training Emissions of Hugging Face CV & Multi-Modal Models (5,000+ Downloads)

entries, and TPUs are handled via dedicated TPU families. Only when no reliable family can be inferred do we use an A100-class default for GPUs or TPU V3 for TPUs, keeping assumptions conservative and internally consistent. Peak compute throughput (TFLOPs/s) and average power consumption are tabulated for major GPU and TPU families under FP16/BF16 tensor-core settings. Custom mappings standardize diverse naming conventions (e.g., "A100 80GB", "TPUv4-8"), while TPU pod descriptors are canonicalized into TPU V2/V3/V4/V5E/V5P. Throughput efficiency is set by accelerator type as shown in Table 6.

**System Overheads.** We include cluster-level overheads beyond accelerator power: (i) an IT overhead factor (20% relative to GPU draw) covering CPU/RAM/NIC usage, (ii) fixed per-node power (250 W), and (iii) per-node network overhead (100 W). A unified PUE of 1.2 accounts for datacenter infrastructure inefficiency.

**Input Integration.** The estimator merges three data sources: (a) GPU/TPU metadata (type, count, nodes, duration), (b) expected FLOPs from scaling estimates or disclosures, and (c) regional emission factors (tCO$_2$/MWh).

**Runtime Attribution.** Two pathways are implemented:

1. **Direct runtime:** If *training hours* are disclosed, emissions are computed directly from reported wall-clock or GPU-hours multiplied by hardware power draw.

2. **Imputed runtime:** If training duration is *not* disclosed but total FLOPs are available, we back-compute runtime as

$$T = \frac{F_{\text{train}}}{\text{PeakTFLOPs} \times R_{\text{eff}} \times N_{\text{acc}}}, \tag{13}$$

   where $F_{\text{train}}$ is expected FLOPs, $R_{\text{eff}}$ is throughput efficiency, and $N_{\text{acc}}$ is accelerator count. This ensures models with only FLOPs disclosure can still be assigned a plausible runtime estimate.

If neither hours nor FLOPs are available, the case is labeled `insufficient`, which is then categorized as a tier 2 or tier 3 model.

*Table 6.* Canonical Accelerator Families Used in Estimation

| Accelerator Family | Peak TFLOPs | Avg. Power (W) | Efficiency |
|---|---|---|---|
| A100 | $3.12 \times 10^{14}$ | 400 | 0.35 |
| A100 80GB | $3.12 \times 10^{14}$ | 400 | 0.35 |
| A100 64GB | $3.12 \times 10^{14}$ | 400 | 0.35 |
| A800 | $3.12 \times 10^{14}$ | 350 | 0.30 |
| H100 | $9.89 \times 10^{14}$ | 600 | 0.45 |
| H200 | $1.00 \times 10^{15}$ | 650 | 0.45 |
| H800 | $8.00 \times 10^{14}$ | 550 | 0.40 |
| V100 | $1.25 \times 10^{14}$ | 300 | 0.25 |
| T4 | $6.5 \times 10^{13}$ | 70 | 0.20 |
| L4 | $1.20 \times 10^{14}$ | 75 | 0.25 |
| A40 | $3.00 \times 10^{14}$ | 300 | 0.25 |
| A30 | $1.65 \times 10^{14}$ | 300 | 0.25 |
| RTX 6000 ADA | $1.45 \times 10^{14}$ | 300 | 0.25 |
| MI250X | $3.83 \times 10^{14}$ | 560 | 0.30 |
| MI300X | $1.20 \times 10^{15}$ | 750 | 0.40 |
| TPU V2 | $4.5 \times 10^{13}$ | 120 | 0.25 |
| TPU V3 | $1.23 \times 10^{14}$ | 187 | 0.35 |
| TPU V4 | $2.75 \times 10^{14}$ | 220 | 0.45 |
| TPU V5E | $8.0 \times 10^{13}$ | 120 | 0.35 |
| TPU V5P | $2.90 \times 10^{14}$ | 280 | 0.45 |

**Emission Calculation.** Total energy consumption is given by

$$MWh = \Big( P_{\text{acc}} \cdot N_{\text{acc}} \cdot T \ + \ \text{IT overhead} \ + \ \text{node/network fixed} \Big) \times \text{PUE}, \tag{14}$$

where $P_{\text{acc}}$ is average power per accelerator, $N_{\text{acc}}$ the accelerator count, and $T$ the effective training duration (hours). Multiplying by the regional emission factor yields emissions in tCO$_2$e.

### E.2. Emission Estimation with Training Flops for tier 2 models.

We establish a log–log regression between model training FLOPs, regional emission factors, and hardware accelerator families:

$$\log(E_i) \ = \ \beta_0 \ + \ \beta_1 \log(F_i) \ + \ \beta_2 \log(\text{EF}_i) \ + \ \sum_k \gamma_k \, \mathbf{1}\{\text{acc}_i = k\} \ + \ \varepsilon_i, \tag{15}$$

where $E_i$ denotes the training emissions (tCO$_2$e), $F_i$ the expected FLOPs, EF$_i$ the grid emission factor (tCO$_2$/MWh) in the model training region, and $\mathbf{1}\{\text{acc}_i = k\}$ an indicator for accelerator family $k$. The regression yields a robust elasticity of $\beta_1 \approx 0.83$ for FLOPs, and $\beta_2 \approx 0.85$ for grid emission factors, while hardware differences are captured by the categorical terms $\gamma_k$.

Thus, the approximation logic can be expressed as

$$E_i \ \approx \ C \cdot F_i^{0.83} \cdot \text{EF}_i^{0.85} \cdot \delta(\text{acc}_i), \tag{16}$$

where $C = \exp(\beta_0)$ is a constant and $\delta(\text{acc}_i)$ is a multiplicative adjustment depending on the accelerator family.

### E.3. Emission Estimation with Parameters for tier 3 models.

This parameter-based regression is used as a fallback for Tier-3 models, where no additional information is available to support FLOPs-based estimation. We therefore distinguish two cases. If a model provides a usable parameter count, we

*Table 7.* OLS regression of log-emissions on FLOPs, grid emission factors, and hardware dummies. Robust (HC3) standard errors in parentheses.

| Variable | Coefficient | Std. Error |
|---|---|---|
| Intercept | $-39.252^{***}$ | (1.685) |
| $\log(F)$ | $0.829^{***}$ | (0.034) |
| $\log(\text{EF})$ | $0.847^{**}$ | (0.362) |
| acc[T.H-family] | $-0.827^{**}$ | (0.331) |
| acc[T.Others] | $0.629$ | (0.389) |

$^{***}p < 0.01, ^{**}p < 0.05, ^{*}p < 0.1$

estimate its emissions using a parameter-based regression. If no usable training-related metadata, including parameter counts, is available, we impute emissions using the average emission level of comparable models in the same category.

For a parameter-based regression, we establish a log–log regression between model parameter counts, regional emission factors, and model subtype categories:

$$\log(E_i) = \beta_0 + \beta_1 \log(P_i) + \beta_2 \log(\text{EF}_i) + \gamma \mathbf{1}\{\text{subtype}_i = \text{finetune}\} + \varepsilon_i, \tag{17}$$

where $E_i$ denotes the training emissions (tCO$_2$e), $P_i$ the parameter count of the model, $\text{EF}_i$ the grid emission factor (tCO$_2$/MWh), and $\mathbf{1}\{\text{subtype}_i = \text{instruct}\}$ an indicator for instruction-tuned models.

The regression indicates an elasticity of $\beta_1 \approx 1.45$ with respect to parameters, while the effect of grid emission factors is smaller and statistically insignificant. Instruction-tuned variants show systematically lower emissions compared to base models.

Thus, the approximation logic can be expressed as

$$E_i \approx C \cdot P_i^{1.45} \cdot \text{EF}_i^{0.34} \cdot \delta(\text{subtype}_i), \tag{18}$$

where $C = \exp(\beta_0)$ is a constant and $\delta(\text{subtype}_i)$ is a multiplicative adjustment depending on whether the model is instruction-tuned.

*Table 8.* OLS regression of log-emissions on parameter counts, grid emission factors, and subtype dummies. Standard errors in parentheses.

| Variable | Coefficient | Std. Error |
|---|---|---|
| Intercept | $-32.127^{***}$ | (1.139) |
| $\log(P)$ | $1.451^{***}$ | (0.054) |
| $\log(\text{EF})$ | $0.343$ | (0.250) |
| subtype[T.instruct] | $-1.001^{***}$ | (0.297) |

$^{***}p < 0.01, ^{**}p < 0.05, ^{*}p < 0.1$

## F. Uncertainty Propagation for Aggregate Industry-Level Emissions Estimations

Our framework categorizes training-emission estimates into three levels. Each introduces uncertainty from different sources. This section details the origin and nature of these uncertainties and how they propagate into the final emission estimates.

**Tier-1: Fully or Partially Disclosed Training Metadata**  Tier-1 models provide the most reliable information and fall into two subcategories.

**(a) Direct disclosure.** Some models report one or more of electricity consumption (MWh) or CO$_2$e emissions; GPU/TPU-hours; explicit accelerator type and count; training region or datacenter provider. In these cases, emissions follow the standard power–time formulation

$$E_{\text{T1}} \approx \text{MWh} \times EF_{\text{region}}, \tag{19}$$

with uncertainty dominated only by reporting granularity (rounding, coarse region labels).

**(b) High-confidence FLOP-based Tier-1.** For other Tier-1 models, total training FLOPs are disclosed or recoverable with high fidelity (e.g., from official technical reports), and emissions are computed as

$$E_{\text{T1}} \approx F_{\text{train}}^{\text{total}} \ \times \ K_{\text{eff}} \ \times \ EF_{\text{region}}. \tag{20}$$

Here, $K_{\text{eff}}$ represents the effective electricity consumption per unit of compute:

$$K_{\text{eff}} \ = \ \frac{P_{\text{GPU}} \times A_{\text{time}}}{\theta_{\text{GPU}} \times \text{peakTFLOPS}}, \tag{21}$$

where $P_{\text{GPU}}$ is the average power draw, $\theta_{\text{GPU}}$ is the achieved utilization efficiency, and $A_{\text{time}}$ is a runtime amplification factor capturing communication, I/O, and other overheads.

Uncertainty therefore propagates primarily through small variations in $\theta_{\text{GPU}}$, $A_{\text{time}}$, and regional emission factors. Because both $F_{\text{train}}^{\text{total}}$ and the hardware family are well constrained, Tier-1 FLOP-based estimates also exhibit low uncertainty.

**Tier-2: FLOPs Known, Hardware and Runtime Partially Missing**   Tier-2 models disclose (or allow reconstruction of) the total training FLOPs, but lack full hardware/runtime information. Emissions are therefore computed as

$$E_{\text{T2}} \approx F_{\text{train}}^{\text{total}} \, K_{\text{eff}} \, EF_{\text{region}}, \tag{22}$$

where $K_{\text{eff}}$ groups accelerator throughput, datacenter amplification, PUE, and average power.

Tier-2 uncertainty thus arises from:

1. Imputed hardware family (A100/A800/H100/TPU/AMD),

2. Throughput/efficiency variance in $\theta_{\text{GPU}}$ across implementations and parallelism setups,

3. Datacenter amplification uncertainty ($A_{\text{time}}$),

4. Regional EF uncertainty due to missing or ambiguous geography.

Because FLOPs is known while $K_{\text{eff}}$ and $EF_{\text{region}}$ are imputed, Tier-2 inherits moderate uncertainty.

**Tier-3: Neither FLOPs Nor Runtime Disclosed**   Tier-3 models require the heaviest imputation. Total FLOPs must be estimated from model parameters via a scaling-law style approximation:

$$F_{\text{train}}^{\text{total}} \approx c \, N_{\text{params}}, \tag{23}$$

where the coefficient $c$ implicitly absorbs typical choices of token counts, training stages (pretraining, SFT, RLHF), number of epochs, and curriculum details for a given family of models.

Emissions then follow:

$$E_{\text{T3}} \approx (c \, N_{\text{params}}) \, K_{\text{eff}} \, EF_{\text{region}}. \tag{24}$$

Major sources of Tier-3 uncertainty include:

1. Scaling-law coefficient variance (the proportionality constant $c$ is architecture- and corpus-specific and absorbs variation in effective token counts and training stages);

2. Hardware inference as in Tier-2 (accelerator family, utilization, and datacenter amplification folded into $K_{\text{eff}}$);

3. Regional EF uncertainty when geography is missing or coarse;

4. Compounded multiplicative propagation across $F_{\text{train}}^{\text{total}}$, $K_{\text{eff}}$, and $EF_{\text{region}}$.

Since both $F_{\text{train}}^{\text{total}}$ and $K_{\text{eff}}$ must be imputed, and each term enters multiplicatively, Tier-3 accumulates the largest theoretical error. Plugging representative relative uncertainties as shown in Table 9 into Eq. 6 yields

$$\frac{\Delta E}{E} \approx \sqrt{\left(\frac{\Delta F}{F}\right)^2 + \left(\frac{\Delta K_{\text{eff}}}{K_{\text{eff}}}\right)^2 + \left(\frac{\Delta EF}{EF}\right)^2} \sim 0.9\text{–}1.5, \tag{25}$$

corresponding to an implied Tier-3 uncertainty range of $\pm(90\text{–}150)\%$, i.e., roughly **2–3**$\times$ variation for typical models.

*Table 9.* Typical relative-uncertainty ranges for multiplicative factors in Eqs. (3)–(6).

| Quantity | Symbol | Typical Relative Error ($\Delta x/x$) |
|---|---|---|
| Total training FLOPs (Tier-1/2) | $\Delta F/F$ | 0.05–0.15 |
| Total training FLOPs (Tier-3 proxy $cN_{\text{params}}$) | $\Delta F/F$ | 0.60–0.80 |
| GPU average power draw | $\Delta P/P$ | 0.05–0.10 |
| Utilization efficiency | $\Delta\theta/\theta$ | 0.10–0.25 |
| Runtime amplification factor | $\Delta A_{\text{time}}/A_{\text{time}}$ | 0.10–0.20 |
| Regional emission factor | $\Delta EF/EF$ | 0.10–0.20 |

**Summary of Error Sources and Expected Magnitudes**

- **Tier-1 (low)**: $\pm 5\text{–}15\%$ (direct or high-confidence FLOPs-based; minimal imputation).

- **Tier-2 (moderate)**: $\pm 40\text{–}70\%$ (hardware, efficiency, and EF imputation; FLOPs accurate).

- **Tier-3 (high)**: $\pm 90\text{–}150\%$ (both FLOPs proxy $cN_{\text{params}}$ and hardware/datacenter effects imputed; multiplicative compounding).

These theoretical ranges follow directly from the multiplicative structure in Eqs. (3)–(5), the first-order propagation rule (Eq. (6)), and representative relative uncertainties as shown in Table 9. The theoretical ranges are also consistent with our pseudo-missingness experiments in Appendix H.

**Aggregate Uncertainty.** Considering the expected uncertainty of each tier (Tier 1: 10%, Tier 2: 55%, Tier 3: 120%) by their respective emissions proportions (Tier 1: 33%, Tier 2: 60%, Tier 3: 7%), we yield an aggregate-level uncertainty of approximately $\pm 40\%$. Thus, our estimates indicate that, as of August 2025, training 5,227 models with more than 5,000 downloads has resulted in cumulative emissions of approximately $6.0 \times 10^4$ tCO$_2$e with an uncertainty of $\pm 2.4 \times 10^4$ tCO$_2$e.

**Significant-digit rules.** Reporting results follow standard significant-digit rules: aggregate emissions are given with at most two significant digits, and uncertainty intervals with one significant digit. For ATCI, we apply the same significant–digit principles. Because ATCI is a ratio of two quantities with comparable relative uncertainty (emissions and FLOPs). Accordingly, ATCI values are reported with one to two significant digits, matching the precision justified by the input factors and the error structure in Eq. (6).

# G. Variance-Based Uncertainty Decomposition

This section evaluates the uncertainty from the perspectives of FLOPs estimation, hardware/supercomputing efficiency, and grid emission factors. We model training emissions as:

$$E \approx \text{FLOPs} \times \text{EF} \times K, \tag{26}$$

where EF is the regional emission factor and $K$ absorbs hardware efficiency, runtime, and PUE effects. For each model, we infer $K_{\text{eff}} = E/(\text{FLOPs} \times \text{EF})$ and perform a variance-based sensitivity analysis with realistic perturbations:

- FLOPs: $\pm 30\%$ uncertainty,

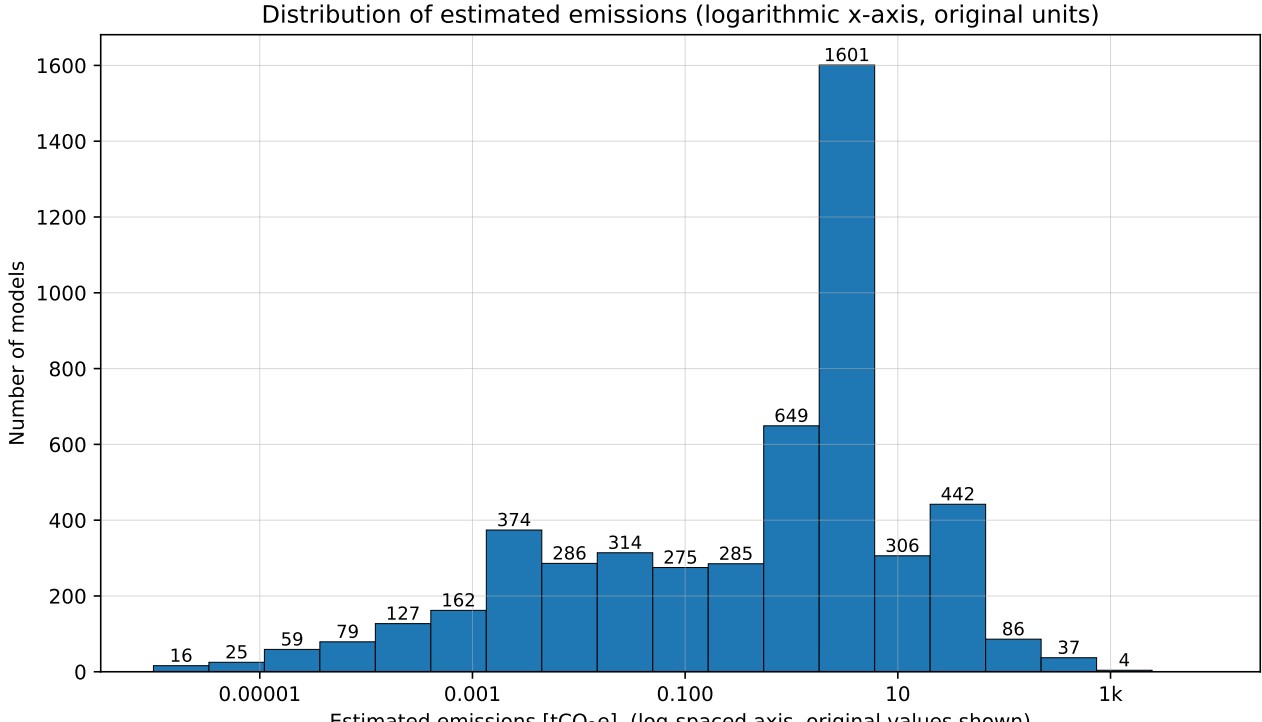

*Figure 9.* Distribution of Estimated Emissions of Hugging Face Models (5,000+ Downloads)

- Hardware/runtime/PUE: $\pm 20\%$,

- Grid EF: $\pm 10\%$.

Using 1,000 Monte Carlo samples per factor, we estimate each source's contribution to $\mathrm{Var}(E)$. The global contributions averaged across all models are:

*Table 10.* Variance-based uncertainty decomposition.

| Uncertainty Source | Variance Share |
| --- | --- |
| FLOPs estimation | 66% |
| Hardware/runtime/PUE ($K$) | 27% |
| Grid emission factor (EF) | 7% |

FLOPs estimation constitutes the dominant uncertainty driver, while EF accounts for only a small fraction. Based on variance decomposition across estimation components, FLOPs estimation contributes $\sim$66% of overall uncertainty, hardware assumptions $\sim$27%, and grid emission factors $\sim$7%. These results demonstrate that uncertainty arises primarily from ecosystem-wide metadata sparsity rather than methodological limitations.

## H. Pseudo-Missingness Experiment for Tier 2 and Tier 3 Uncertainty

To explicitly quantify the uncertainty introduced by Tier 2 and Tier 3 estimation, we conduct a **pseudo-missingness experiment** that closely aligns with real metadata disclosure patterns observed on Hugging Face.

**Ground-truth selection.** We use **all Tier 1 models** as high-confidence ground truth, including those with direct energy disclosure or those with complete metadata (training hardware and training GPU hours). To avoid numerical instability in relative errors, we remove only trivial-emission cases, eliminating numerical artifacts while preserving essentially all meaningful Tier 1 models.

**Constructing pseudo Tier 2 / Tier 3 samples.**    We randomly sample 70% of Tier 1 models and **artificially mask metadata** to simulate realistic missingness:

- **Pseudo Tier 2:** retain FLOPs, emission factor, and GPU family; mask hardware type, runtime, and direct/disclosed energy.

- **Pseudo Tier 3:** further remove FLOPs, leaving only parameter count, emission factor, and GPU family.

These masked models are re-evaluated using the **exact Tier 2 and Tier 3 regression pipelines** described in the paper. Predicted emissions are compared with Tier 1 ground truth using absolute error (AE) and relative error (RE). Results are shown in Table 11.

*Table 11.* Pseudo-missingness experiment results for Tier 2 and Tier 3 uncertainty.

| Pseudo Tier | n | MAE (tCO$_2$e) | Median RE | P90 RE |
|---|---|---|---|---|
| Tier 2 (FLOPs-based) | 312 | 61.62 | **0.57** | **1.20** |
| Tier 3 (Params-based) | 123 | 111.42 | **0.99** | **1.92** |

- **Tier 2 estimates remain highly stable:** median RE $\approx 0.57$; 90% of predictions exhibiting $\sim 1.2\times$ relative error.

- **Tier 3 remains informative despite minimal metadata:** median RE $\approx 0.99$; 90% within $\sim 2\times$ relative error.

Median RE summarizes the **typical multiplicative deviation** introduced when metadata is partially or severely missing. For example, a Median RE of $0.57$ indicates that half of the reconstructed emissions differ from the Tier 1 ground truth by no more than $57\%$, while the remaining half may exhibit larger deviations.

In this context, Median RE captures how much accuracy can be preserved when Tier 1 quality metadata is downsampled to the more realistic, incomplete metadata available under Tier 2 or Tier 3 conditions. A low Median RE for pseudo Tier 2 suggests that FLOPs and emission factors alone are sufficient to retain a substantial fraction of estimation fidelity. These results show that Tier 2 and Tier 3 estimates are not exact but remain **predictive at the order-of-magnitude level under realistic missingness patterns**.

**Additional uncertainty mitigation mechanisms.**    To constrain uncertainty, our framework incorporates:

- architecture-based FLOPs derivation and runtime backsolving with bounded parameter ranges,

- GPU-family regression calibrated on Tier 1 ground-truth models,

- variant deduplication to avoid double-counting mirrors or lightweight derivatives,

Together, these mechanisms ensure that Tier 2 and Tier 3 predictions remain anchored to validated Tier 1 models and provide stable, interpretable estimates across the open-source model ecosystem.

