# OpenReview forum: "Hugging Carbon: Quantifying the Training Carbon Emissions of AI Models at Scale"
_ICML.cc/2026/Conference — ICML 2026 regular_

### Official Review · Reviewer_7SJM · 2026-03-08

**Soundness:** 3
**Presentation:** 3
**Significance:** 3
**Originality:** 3
**Overall Recommendation:** 5
**Confidence:** 4

**Summary:**

The authors propose a framework for making aggregate estimates of carbon emissions from training AI models. They consider models hosted on Hugging Face and calculate carbon emissions estimate using variable, often incomplete, amounts of information reported in the repositories. When information is incomplete, estimates are grounded on a smaller set of 390 NLP and 352 CV/multimodal repos that provide CO2 emissions with enough detailed information to cross-check the reported value with -- estimates are made with a FLOPs-based approach when possible, with assumptions made when necessary about details like compute region.

**Compliance With Llm Reviewing Policy:**

Affirmed.

**Final Justification:**

My assessment of this work was already positive due to its timeliness and overall principled analogy. Increasing my score (4 -> 5) after discussion, which adequately addressed my remaining concerns which were about assumptions about MoE models' FLOPs. The problem at hand inherently requires the researchers to make some assumptions and methodological decisions due to inconsistent data availability, but the authors have made defensible and sound decisions, and they appropriately scope their experimentation and claims

**Key Questions For Authors:**

1. Are there any plans to release a dataset of the models/repos that the analyses presented are on? I feel this could strengthen the contribution of the work
2. In 4.3, it is mentioned that 2,422 repositories have a `co2_eq_emissions` field, but that only 292 models were analyzed. The authors also mention "tier 1" 390 NLP and 352 CV/multimodal repos. What exactly accounts for the discrepancy between these figures? Are the missing 2,130 (or 450) repos/models all excluded due to "unreliable disclosures", numerical instability, and trimming of the highest and lowest values, or are the 292 merely a subset of those remaining after filtering for high quality disclosures? If the latter is correct, how many repositories were excluded? What exactly constitutes  an "unreliable" disclosure or a "numerically unstable" case? (Including the relevant quote below for convenience)
> To evaluate the accuracy of our framework against ground-truth disclosures, we analyze 292 models that publicly report their total training emissions (see Appendix A and G). To ensure robustness, we exclude unreliable disclosures and numerically unstable cases, and adopt a robust trimming procedure to mitigate the impact of heavy-tailed outliers.
3. If the authors were to compare *aggregate* disclosed and estimated values for just the subset of models where co2 emissions was reported, what would the disclosed vs emissions sums look like? Table 2 was illustrative, but it would be helpful to also see a row(s) for aggregate numbers (e.g. one with just the 292, one with all 2,422 with co2_eq_emissions fields, and maybe one with other filtering)

**Limitations:**

yes

**Strengths And Weaknesses:**

Strengths:
1. In contrast to most previous works estimating carbon emissions associated with training AI models, which tend to focus on a single model or family of models and rely on internal telemetry, they aim to establish a methodology for arriving at reasonable carbon emissions estimates at scale using often-incomplete information.
2. In general the paper was well-written with good positioning wrt related work, which makes the contributions clear
3. The authors are transparent about their estimations' uncertainty, which reasonably differs across the 3 tiers of information completeness for the HF models studied

Weaknesses:
1. I am skeptical of some of the assumptions made -- I agree that FLOPs can make sense in aggregate as a proxy or predictor for energy and carbon, but in particular it feels unrealistic to assume, for example, that when training MoEs we should only count FLOPs associated with the active parameters, when the very fact that it is an MoE typically leads to very different assumptions relating to hardware configuration vs a dense model. I would have thought that there would be an additional coefficient beyond just the $c_{arch}$ that is influenced by only the ratio between feed-forward and attention operations
2. Some unclear methodology in 4.3, see questions below
3. Despite the transparency around estimation uncertainty, I felt the relevant information could be better organized given the inherent limitations of the problem (i.e. missing information necessitating many assumptions) -- a visualization could be be helpful to show the effects of different assumptions and the magnitude of the uncertainty


Minor comments:
1. related work published at CHI that similarly attempts to make an aggregate estimate: https://dl.acm.org/doi/full/10.1145/3706598.3714227
2. I would prefer to see "metric tons" spelled out the first time "tons" of CO2 are mentioned -- tCO2e + kg as they are used throughout implies it, but I found it slightly confusing to see "tons" alone as it appears in the abstract
3. The scope of the analysis is limited to open source models hosted on Hugging Face -- though this is mentioned by the authors in the limitations section, the limitation is understated imo. I would prefer to see the limitation stated more explicitly up front, and I don't believe this would significantly weaken the apparent contribution either -- the 5.8x10^4 tCO2e is effectively a lower bound

---

> ### Author Rebuttal · Authors · 2026-03-31
>
> We thank the reviewer for the thoughtful suggestions and positive evaluation. We provide our point-by-point responses below.
>
> > **W1:" MoE typically leads to very different assumptions relating to hardware configuration vs a dense model."**
>
> **A1:**  We agree on the distinct energy characteristics of MoE architectures. Our framework already accounts for the additional computational cost that MoE architectures may introduce. As detailed in Appendix C (lines 771–772), we do so by applying a multiplicative factor to capture the extra overhead of MoE models relative to dense architectures.
>
> We acknowledge this remains an approximation, as full routing and system-level overheads are often not disclosed, which may lead to underestimation. These uncertainties are accounted for in our sensitivity analysis, where hardware/system assumptions contribute 27% of the total variance. While MoE models represent a small portion of the dataset (3.5%), any remaining bias is unlikely to materially affect aggregate results. We will clarify and refine this in the revision.
>
> > **W3. "visualization could be be helpful to show the effects of different assumptions and the magnitude of the uncertainty"**
>
> **A3:** Thanks for the suggestions. In the revision, we will add a sensitivity decomposition plot with different assumptions that shows how each assumption contributes to total uncertainty.
>
> > **W4. "related work published at CHI that similarly attempts to make an aggregate estimate."**
>
> **W4:** We thank the reviewer for pointing out this relevant work and will include it in the revision. The related work focuses on a different stage, namely usage-phase emissions. Specifically, it estimates emissions from generative AI usage in HCI research by analyzing how researchers interact with AI systems, such as through user studies. In contrast, our work targets training-phase emissions at the model level, deriving emissions from training FLOPs, hardware efficiency, and system factors.
>
> > **W5. "I would prefer to see "metric tons" "**
>
> **A5:** We will spell out "metric tons" on first mention for clarity in the revised version.
>
> > **W6."The scope of the analysis is limited to open source models hosted on Hugging Face... I would prefer to see the limitation stated more explicitly up front, and I don't believe this would significant."**
>
> **A6:** We will revise the introduction to make this scope more explicit upfront. We also thank the reviewer for highlighting this point. More broadly, estimating carbon emissions across the AI ecosystem, including both open-source and closed-source models, is part of our long-term vision. We appreciate the reviewer’s recognition of the importance of open-source emissions estimation.
>
> > **Q1. "Are there any plans to release a dataset of the models/repos that the analyses presented are on? I feel this could strengthen the contribution of the work "**
>
> **QA1:** Yes, we will publicly release both the curated dataset of models/repositories used in our analysis and the corresponding analysis code.
>
> > **W2,Q2. "What exactly accounts for the discrepancy between these figures? ...how many repositories were excluded? What exactly constitutes an "unreliable" disclosure or a "numerically unstable" case?"**
>
> **QA2:** These numbers correspond to different filtering stages and units (repos vs. models).
>
> - **2,422 repos:** all HF repositories with a `co2_eq_emissions` field, without filtering (including all repos downloads >= 0).
> - **Tier 1 repos (390 NLP + 352 CV/MM):** HF repositories with downloads >=5K and relevant energy or emission disclosures, but still heterogeneous in overall data quality.
> - **292 models:** a final high-quality subset used for uncertainty analysis, obtained by applying:
>   1) downloads >5K,
>   2) with a `co2_eq_emissions` field or direct emission disclosures in metadata (energy-only reports are excluded to avoid additional uncertainty from regional emission factor conversion),
>   3) deduplication (removing mirrors/duplicates),
>   4) quality control (excluding unreliable or numerically unstable cases).
>
> For data quality definitions, unreliable disclosures indicate clear inconsistencies (e.g., unit errors, orders-of-magnitude errors, or typos). Numerically unstable means values that yield implausible results under cross-checks (e.g., unrealistically low). We will revise the paper to make this filtering and exclusion more explicit.
>
> > **Q3. "what would the disclosed vs emissions sums look like? "**
>
> **QA3:** We thank the reviewer for this helpful suggestion. We will include aggregate statistics in Table 2 in the revision.
>
> - For the 292 high-quality Tier 1 models, total disclosed emissions are 17,972.61 metric tons, compared to 12,640.14 metric tons from our estimates.
> - For the illustrative models in Table 2, disclosed emissions are 11,320.02 metric tons, versus 9,707.07 metric tons estimated.

---

> > ### Author Rebuttal · Reviewer_7SJM · 2026-04-02
> >
> > Thank you for the thorough response, which addresses most of my concerns. I have also read other reviewers' points and my assessment of the work remains no less positive overall. In particular, as a researcher who has published and reviewed work that is similar in spirit, I would like to acknowledge the inherent challenge the authors have undertaken; they have made overall thoughtful decisions, and my remaining concerns should not be seen as an indictment against the general soundness of their work.
> >
> > > Our framework already accounts for the additional computational cost that MoE architectures may introduce. As detailed in Appendix C (lines 771–772), we do so by applying a multiplicative factor to capture the extra overhead of MoE models relative to dense architectures.
> >
> > Lines 771-772 refer to a small (1-1.05x) "routing overhead factor" applied to the effective parameter count that affects the FLOPs estimation, rather than the **hardware** considerations that I specifically mentioned being concerned about when it came to energy consumption estimations. Concretely, beyond memory and communication overhead, one either needs a different hardware configuration or different assumptions about hardware utilization vs when training a dense model. To my understanding a 1.05x correction factor is unrealistic
> >
> > > hardware/system assumptions contribute 27% of the total variance
> > > MoE models represent a small portion of the dataset (3.5%)
> >
> > It seems that hardware/system assumptions are likely *understated* for MoEs if the vast majority of the dataset is dense models -- I do not necessarily question the aggregate results reported in the present study, but I have reservations about handwaving assumptions around such a fundamentally different and prominent class of models that are likely heavily represented among closed models.
> >
> > > downloads >5K,
> >
> > > with a co2_eq_emissions field or direct emission disclosures in metadata (energy-only reports are excluded to avoid > additional uncertainty from regional emission factor conversion),
> >
> > > deduplication (removing mirrors/duplicates),
> >
> > > quality control (excluding unreliable or numerically unstable cases).
> >
> > This helps a lot! Would you be able to provide numbers for how many are excluded due to each of these? (Also specifically how many were found to be "unreliable" vs "numerically unstable.")
> >
> > > Numerically unstable means values that yield implausible results under cross-checks (e.g., unrealistically low).
> >
> > Would you mind walking through a concrete example or two?

---

> > > ### Author Response · Authors · 2026-04-03
> > >
> > > Thank you for the reviewer’s thoughtful comments and constructive feedback. We especially appreciate the recognition of the challenges of this problem and the overall soundness of our work.
> > >
> > > > **Q1, Q2. Hardware configuration or assumptions for MOE**
> > >
> > > We thank the reviewer for raising this important point. We agree that MoE models introduce non-negligible system-level overheads beyond FLOPs. Although our current formulation uses a multiplicative factor originally attributed to routing, we agree that this should instead capture the combined effect beyond memory and communication overhead. We will therefore revise our formulation by updating the unified correction coefficient.
> > >
> > > To better ground this coefficient, we reviewed available evidence on large-scale MoE training:
> > >
> > > - [R1] All-to-All communication accounts for 34.1% of step time, up to 74.9% within a single MoE layer.
> > > - [R2] Communication accounts for 32% of total training time and 43.6% of forward time, with MFU dropping from 32.5% to 27.9%.
> > > - [R3] identifies communication and activation memory as dominant bottlenecks, with utilization dropping to <10% of peak FLOPs in some cases.
> > > - [R4] Communication can exceed 50% of total training time without careful optimization.
> > > - [R5] Meta empirical measurement (Appendix D): 160 vs 115 TFLOPs/GPU (dense vs MoE) on A100, corresponding to ~1.39× runtime/energy amplification.
> > >
> > > Based on [R5] and the broader evidence above, these findings suggest that our original estimate is conservative, and that a more realistic correction factor is 1.4×-2.0×, depending on system configuration.
> > >
> > > We will revise the correction factor within this empirically supported range and explicitly discuss its uncertainty. Finally, although MoE models constitute a small portion of our dataset (3.5%), we will clarify this limitation and improve their system-level modeling in the final version.
> > >
> > > - [R1] Accelerating Distributed MoE Training and Inference with Lina
> > > - [R2] MegaScale-MoE: Large-Scale Communication-Efficient Training of Mixture-of-Experts Models in Production
> > > - [R3] X-MoE: Enabling Scalable Training for Emerging Mixture-of-Experts Architectures on HPC Platforms
> > > - [R4] Optimizing Communication for Mixture-of-Experts Training with Hybrid Expert Parallelism
> > > - [R5] Efficient Large Scale Language Modeling with Mixtures of Experts
> > >
> > >
> > > > **Q3. Would you be able to provide numbers for how many are excluded due to each of these?**
> > >   1) **downloads >5K**: In Hugging Face, there are 6,638 repositories with downloads ≥5K as of August 2025, forming the initial candidate pool for our analysis.
> > >   2) **deduplication**: After deduplication, we retain 5,234 repositories (and corresponding models), meaning 1,404 repositories are removed.
> > >   3) **with `co2_eq_emissions` or direct emission disclosures**: This leaves 338 models with directly disclosed emissions information, which means 4,896 models are removed.
> > >   4) **quality control**: After manual verification, we retain 292 models, which means we removed 46 models in total (13 models due to numerically unstable cases, and 33 models due to unreliable cases)
> > >
> > > > **Q4. Numerically unstable: Would you mind walking through a concrete example or two?**
> > >
> > > Here we provide several concrete examples:
> > >
> > > - **unrealistically high**: One example is *nvidia/Cosmos-Reason1-7B*, for which a disclosure reported 5,380 metric ton emissions with metadata disclosing 3.26 × 10^21 training FLOPs associated with training.
> > >
> > > Given its 7B scale and training flops, this implies an energy intensity far above comparable models under similar hardware assumptions. It is nearly ten times the estimated emissions of GPT-3 (175B Parameters, 552.1 metric tons emissions, training flops: 3.14 × 10^23) [R6]. We therefore excluded it from the disclosed value, considering it numerically unstable.
> > >
> > > Importantly, we found that this value is no longer present in the current repository now, suggesting that the original disclosure has been revised or removed. The earlier version can still be accessed via the AWS Marketplace listing and the Hugging Face mirror repo (unsloth/Cosmos-Reason1-7B).
> > >
> > > [R6] Patterson, David, et al. "Carbon emissions and large neural network training." arXiv preprint arXiv:2104.10350 (2021).
> > >
> > > - **unrealistically low**: The `co2_eq_emissions` field of  *KoalaAI/Text-Moderation* shows that it only emitted 0.04g CO₂ eq emissions during training. Given that the model is based on a DeBERTa-scale architecture, the disclosed value is only a few seconds of GPU execution when converted to energy consumption.
> > >
> > > We can perform a simple back-of-the-envelope check: Assuming a typical carbon intensity of ~0.4 tCO₂e/MWh, 0.04 g CO₂e corresponds to only ~0.0001 kWh of energy. This is equivalent to nearly 1 second of GPU runtime under typical accelerator power (e.g., A100 at ~400W), which is inconsistent with any full training or fine-tuning process. As a result, we removed such unrealistically low value to avoid systematic underestimation.

---

### Official Review · Reviewer_GpiJ · 2026-03-12

**Soundness:** 2
**Presentation:** 3
**Significance:** 3
**Originality:** 3
**Overall Recommendation:** 3
**Confidence:** 4

**Summary:**

This work looks to produce a large-scale audit of the energy usage and carbon emissions of open-weight training runs. Based on a subset of models on Hugging Face, they propose a methodology that attempts to deal with discrepancies across reporting of these models and group them into three tiers, each with increasing uncertainty. While the final uncertainty of their estimates remains high, they frame their work as a first step in standardising audit and energy consumption estimates in AI model training.

**Compliance With Llm Reviewing Policy:**

Affirmed.

**Final Justification:**

I have had a back and forth with the authors, and while I agree that this work is valuable and timely, I still hold many concerns about the utility of this work, especially with the extent of the estimates that have to do with the ancillary compute used during training and the authors' claims about that being too difficult to estimate and double-counting. I hope they can address these adequately in the future. I have upped my confidence to 4 and my originality to 3.

**Key Questions For Authors:**

- Why didn't you cite the Stochastic Parrots paper?

**Limitations:**

- See my notes on Soundness.

**Strengths And Weaknesses:**

Presentation: Figure 1 could be made more intuitive by adding more details. For example, what is a mid-sized EU country? For the most part, presentation quality is adequate.

Soundness:
(1) I'm not sure that considering only models with >5,000 downloads is a good proxy of the wider industry. This excludes big models trained on many GPUs that, for whatever reason, fail to gain traction. Also, what is the distribution at the tail end? It would seem to me that the downloads would follow a power distribution, but it does not follow that the compute requirements would also follow this distribution.
(2) Also, I believe the authors mention that most compute demands will end up being during the inference phase. As a result, I find it problematic that they are not doing any deeper analysis here. I think, given the limited scope that the authors established, this work seems adequately sound.
(3) I would imagine that most of the energy consumption during training is done within closed-weight labs. As such, looking solely at open-weight models seems quite limiting to me.
(4) This work does not adequately concern itself with ancillary training work, such as experiments and ablations, and chooses to solely focus on the final full training run. This seems pretty limiting to me.
(5) There is a disconnect between FLOPS and actual energy usage. For example, moving data to and from memory to the compute core is energy-intensive, but looking only at FLOPS would underrepresent this energy usage.
(6) Given the limitations I've described above, I think your real error bars are much wider than you claim. I will concede that getting this data may be difficult or impossible, but I feel the authors should have given deeper consideration to this.
(7) I think this work is certainly timely, but given these concerns, I'm on the fence on whether it could be publishable in its current state.

Significance: I believe the problem this paper attempts to address is quite important, but I'm not sure they've done the best job in addressing it. See my Soundness notes.

Originality: While not fully original, I think they tackle a valuable issue.

---

> ### Author Rebuttal · Authors · 2026-03-31
>
> We thank the reviewer for the valuable suggestions and for recognizing the timeliness of our work.
>
> >W1:  "considering only models with >5,000 downloads"
>
> **A1**: Our framework is scalable. We focus on ≥5K-download models to ensure data quality, validity, and observability.
> #### Table 1. Attribution Feasibility
> | Bin        | Missing README | Missing Config | Missing Paper or Tech Report |
> |------------|----------------|----------------|----------------|
> | <5K        | 36%          | 48%          | 79%          |
> | [5k,20k)  | 11%          | 21%          | 53%          |
> | [20k,50k) | 6%           | 14%          | 40%          |
> | 50k+       | 4%           | 8%           | 36%          |
>
> - Low-download models exhibit severe metadata sparsity and structural noise. Under our attribution criteria (complete README, config, weights, and training information), we observe a clear break around 5K downloads.
>
> - Beyond missing data, low-download repositories include many non-representative cases (e.g., checkpoints, duplicates, quantized variants), which are difficult to filter at scale and can introduce bias, especially for Tier 2/3 estimation. Even within the ≥5K subset, we apply additional deduplication and manual filtering.
>
> - Models with ≥5K downloads provide a higher signal-to-noise proxy: they are typically better documented, community-validated, and more representative of real-world usage.
>
> The 5K threshold is therefore empirically motivated rather than arbitrary. It serves as a first-stage filter to ensure data quality and enable robust large-scale estimation. We will clarify this motivation in the revision.
>
> >W2: "compute demands during the inference phase."
>
> **A2**: We agree that inference is important, but estimating aggregate inference emissions requires a different problem setting and new data. The key challenge is the lack of deployment-side usage data (e.g., number of calls and tokens processed). More data and estimation methodologies are needed. This is therefore not a missing analysis, but beyond the scope of this work. Estimating training emissions is a necessary first step, and we are still working to collect data and estimate inference emissions; this is a very important part of our future work.
>
> >W3: "energy consumption within closed-weight labs."
>
> **A3**: This work aims to provide a robust estimation framework for settings with partial information. Open-source models serve as an observable testbed to link verified data (Tier 1) with sparse cases (Tiers 2/3), which is necessary before extending to opaque, closed-source settings. Our methodology can be applied once energy-related information from closed-source models becomes available; however, such disclosures are limited, making systematic analysis of closed-source models infeasible.
>
> The open-source ecosystem provides the only auditable basis for carbon accounting. Our work should be viewed not as prioritizing open-source models, but as establishing an empirical foundation that can extend to broader settings when more transparency becomes available.
>
> > W4: "concern itself with ancillary training work."
>
> **A4:** We thank the reviewer for this important point.  We will further clarify this boundary in the revision.
>
> First, we estimate emissions at the level of individual released models, rather than the full project-level R&D lifecycle. Ancillary training produces multiple intermediate variants, and attributing all such costs to each release would lead to systematic double-counting. We therefore focus on the realized cost of each published instance.
>
> Second, this boundary aligns with current reporting practices. Tier 1 disclosures typically reflect finalized training runs or released models, rather than full experimental lifecycles. Our framework is designed to be consistent with these disclosures.
>
> >W5: "disconnect between FLOPS and actual energy usage."
>
> **A5:** FLOPs is not used in isolation: it represents compute demand, while energy is determined by system-level factors (Eq. 4), including (P_GPU / θ_GPU), capturing hardware efficiency and memory/I/O effects. Non-compute components are thus reflected in aggregate. Empirically, emissions scale positively with FLOPs (Fig. 2), with a sub-linear exponent (0.83) indicating efficiency gains and system-level effects.
>
> >W6: "real error bars are much wider."
>
> **A6:** As discussed in our response to W4 - W5, the uncertainty sources highlighted by the reviewer (the FLOPs-energy gap and ancillary training) are not ignored, but are either incorporated into our modeling or fall outside the defined accounting boundary of our framework. We also conduct an uncertainty analysis to evaluate the sensitivity of emissions estimates to key variables. This allows us to characterize both potential underestimation and overestimation.
>
> >  Q1, W0
>
> **QA1:** We will incorporate a citation to the Stochastic Parrots paper. In addition, we will revise Fig. 1 to clarify the term “mid-sized EU country” (e.g., Croatia).

---

> > ### Author Rebuttal · Reviewer_GpiJ · 2026-04-03
> >
> > > We agree that inference is important, but estimating aggregate inference emissions requires a different problem setting and new data. The key challenge is the lack of deployment-side usage data (e.g., number of calls and tokens processed). More data and estimation methodologies are needed. This is therefore not a missing analysis, but beyond the scope of this work. Estimating training emissions is a necessary first step, and we are still working to collect data and estimate inference emissions; this is a very important part of our future work.
> >
> > I think there is more to be done in terms of the carbon footprint of inference, but I am willing to accept it as being left to future work.
> >
> > > First, we estimate emissions at the level of individual released models, rather than the full project-level R&D lifecycle. Ancillary training produces multiple intermediate variants, and attributing all such costs to each release would lead to systematic double-counting. We therefore focus on the realized cost of each published instance.
> >
> > I fail to see how this would lead to "systematic" double-counting. What do you mean by "realized cost"?

---

> > > ### Author Response · Authors · 2026-04-03
> > >
> > > We sincerely thank the reviewer for the thoughtful question and the opportunity to clarify this point. We provide our response below.
> > >
> > > >  **"how this would lead to "systematic" double-counting. What do you mean by "realized cost"?"**
> > >
> > > We thank the reviewer for raising this important point.
> > >
> > > First of all, to clarify terminology, we interpret **ancillary training work** (mentioned in Q4 by the reviewer) as including hyperparameter search, ablation studies, intermediate checkpoints, and failed or exploratory runs, which are typically part of the broader model development process but are not always reflected in the final released model. We hope this interpretation aligns with the reviewer’s definition, and we are happy to further discuss it if needed.
> > >
> > > - Regarding **double-counting**: Model development pipelines involve shared upstream training processes, which are often reused across multiple released models (e.g., fine-tuned or instruction-tuned variants) in Hugging Face. If these shared costs were attributed to each released model, the same underlying training effort would be counted multiple times across repositories, leading to systematic double-counting. Moreover, the mapping between upstream training and released models is not fully observable from public data.
> > >
> > > - Regarding **realized cost**: we define it as the training cost attributable to a finalized, released model instance, based on repository-level information. This provides a non-overlapping unit of analysis without requiring assumptions about how shared upstream costs should be allocated. This also aligns with Hugging Face's reporting practices, where emissions are typically disclosed at the level of the finalized model in the repo.
> > >
> > > Finally, while the ancillary training contributes to the total carbon footprint in the whole R&D lifecycle, incorporating it into our framework would require strong and unverifiable assumptions, because such processes are:
> > > - highly heterogeneous across models and organizations
> > > - not consistently disclosed, and
> > > - not uniquely attributable to a single released model.
> > > Applying a uniform adjustment (e.g., a fixed multiplier for all hyperparameter tuning or experimentation) would introduce higher systematic estimation bias, as the true overhead can vary by orders of magnitude.
> > >
> > > Therefore, we focus on the realized cost of released models as a conservative but robust accounting boundary, which ensures comparability across the Hugging Face ecosystem. This provides a clear and well-defined unit of analysis. We will explicitly clarify this in the revision.

---

### Official Review · Reviewer_ptjp · 2026-03-12

**Soundness:** 3
**Presentation:** 3
**Significance:** 2
**Originality:** 3
**Overall Recommendation:** 3
**Confidence:** 3

**Summary:**

The paper proposes a framework to estimate the carbon emission for training large models leveraging dataset from hugging face. A tiered approach is prosed to fill the missing value. The carbon emission of most popular model is estimated using the proposed framework. Analysis is performed among different models and data types.

**Compliance With Llm Reviewing Policy:**

Affirmed.

**Final Justification:**

The authors address most of my main concerns. The paper shows promise, while I would prefer to maintain my current rating at this stage.

**Key Questions For Authors:**

1. The abstract and conclusion part all mention the training emission with less discussion other parts of results.
2. My main concern is that tracking the carbon emission during the deployment stage might be more meaningful compared to the training stage.

**Limitations:**

yes

**Strengths And Weaknesses:**

Soundness: The paper is technically sound and methods are used appropriate. The author clearly identifies the strength and weakness of the current research work.

Presentation: The paper is clearly written and well-structured and easy to fellow. The paper discusses the previous work and how it differs, specifically previous work mainly analyzes individual models or whole AI cycles, while this study targets a framework for estimating carbon emission of large model training process using the open-source ecosystem on Hugging Face.

Significance: The paper addresses a relevant and timely problem. However, I feel that estimating carbon emissions from inference workloads may be even more important. While training emissions are significant, training is typically unavoidable once a model architecture and dataset are chosen. In contrast, during the deployment stage, practitioners can make decisions about which model to use and where to run inference, potentially optimizing for lower carbon emissions. From this perspective, studying inference-related emissions could provide greater opportunities for practical carbon reduction. Nevertheless, the topic addressed in this paper remains relevant and valuable. This work could be linked to other research areas such as energy sector and climate research.

Originality: The paper proposes framework for estimating carbon emission of large model training process instead of targeting specific model or the whole AI lifecycles. It would be nice that if the equation is referred from other places would be nice to put the reference.

---

> ### Author Rebuttal · Authors · 2026-03-30
>
> We thank the reviewer for recognizing the soundness and timeliness of our work. We appreciate the discussion regarding the significance of training vs. inference emissions. We would like to clarify and emphasize why **quantifying training emissions is also important and complementary to tracking deployment workloads**:
>
> > W1:"estimating carbon emissions from inference workloads may be even more important"; Q2: "My main concern is that tracking the carbon emission during the deployment stage might be more meaningful compared to the training stage"
>
> **A1:** We agree that understanding the full lifecycle carbon footprint of AI systems is important, and the cumulative emissions from inference during long-term deployment are substantial. However, we argue that **quantifying training emissions is a prerequisite and a fundamentally distinct challenge** for the following reasons:
>
> - Training is **an unavoidable and foundational stage** in the AI lifecycle, making it a concrete and actionable entry point for understanding lifecycle-level environmental impact. It represents the embodied carbon of a model instance, which is a concentrated, irreversible investment incurred before a single query is served.
>
> - **Addressing the transparency gap at scale.** At the same time, training emissions remain largely unreported and opaque, especially across thousands of open-source models on platforms such as Hugging Face. By leveraging the HF large-scale ecosystem, our work provides the first systematic effort to quantify training emissions at scale, addressing this critical transparency gap and enabling meaningful comparison across models.
>
> - **Irreversible and System-Critical Decision Point.** Training represents a concentrated and largely irreversible carbon investment, making it a critical decision point in the AI lifecycle. Quantifying these emissions enables both system-level carbon accounting (e.g., power grid planning) and informed early-stage decisions. To support this, we introduce the **ATCI (AI Training Carbon Intensity) metric**, which allows practitioners to evaluate carbon efficiency and assess whether performance gains justify the associated environmental cost.
>
> - **Generalizable and Extensible Framework.** Our framework is built on a FLOPs-based formulation, which can be naturally extended to the deployment stage. By adapting the computation from total training FLOPs to per-token inference FLOPs, the same methodology can be used to estimate inference emissions, if usage data such as the number of inference calls or total processed tokens are available.
>
> We view deployment-stage emissions as an important direction for future work. **Training and inference are distinct and complementary components of the AI carbon lifecycle, and the importance of deployment-stage emissions does not diminish the importance of studying training-stage emissions**. Both are essential problems. However, due to the current lack of large-scale and reliable usage data for deployment stages, studying this stage requires a different set of estimation assumptions and is beyond the scope of the present paper.
>
> > W2: "It would be nice that if the equation is referred from other places would be nice to put the reference."
>
> **A2:**  We thank the reviewer for this suggestion. In the revised manuscript, we will add explicit references to ground our formulation in established literature.
>
> Specifically, for the energy-to-carbon conversion, we will cite standard environmental accounting guidelines, such as the IPCC (2006) framework [1]. For the compute-to-energy relationship, we will reference prior work on estimating energy consumption of NLP deep learning models [2].
>
> We also clarify that one of our contributions lies in Eqs. (4) and (5), which we formulate to provide a decomposition aligned with our regression results and introduce ATCI as a measure of model-level training environmental efficiency.
>
> - [1] Change, I. P. O. C. "2006 IPCC guidelines for national greenhouse gas inventories." Institute for global environmental strategies: Hayama, Kanagawa, Japan (2006).
> - [2] Strubell, Emma, Ananya Ganesh, and Andrew McCallum. "Energy and policy considerations for deep learning in NLP." Proceedings of the 57th annual meeting of the association for computational linguistics. 2019.
>
>
> > Q1: "The abstract and conclusion part all mention the training emission with less discussion other parts of results."
>
> **QA1:** We thank the reviewer for this helpful suggestion. In the revised manuscript, we will improve the abstract and conclusion by summarizing our key empirical findings beyond training emissions. In particular, we will highlight (i) cross-model variation in emissions, (ii) the role of hardware and regional factors, and (iii) uncertainty and robustness results from our analysis.

---

### Official Review · Reviewer_wy2M · 2026-03-15

**Soundness:** 3
**Presentation:** 3
**Significance:** 3
**Originality:** 3
**Overall Recommendation:** 5
**Confidence:** 1

**Summary:**

This paper is a systematic study on estimating the total training carbon emissions of over 5,000 popular open-source AI models hosted on Hugging Face. To overcome the widespread lack of transparent reporting, the authors developed a three-tier estimation methodology that calculates emissions based on the completeness of available metadata. The paper provides the first large-scale empirical baseline for open-source AI carbon accounting.

**Compliance With Llm Reviewing Policy:**

Affirmed.

**Final Justification:**

The author's response addressed my questions.

**Key Questions For Authors:**

See above.

**Limitations:**

See above.

**Strengths And Weaknesses:**

This paper is very far from my domain of expertise. Therefore, I set my confidence to be 1.

One potential weakness in the methodology is the overly simplistic estimation of post-training compute, particularly for RL. While pretraining is traditionally considered the main compute bottleneck, scaling RL is the recent trend, and it's hard to say which one costs more compute between pretraining and RL. Another hidden computational cost that was not considered in this work is hyperparameter tuning.

---

> ### Author Rebuttal · Authors · 2026-03-30
>
> We thank the reviewer for the positive evaluation and constructive feedback. The main concern raised is the hidden cost of post-training in AI carbon accounting. We would like to clarify that **our framework does not overlook these stages**. They are integrated through **analytical formulation, empirical grounding, and statistical inference**.
>
> > **W1. "...the methodology is the overly simplistic estimation of post-training compute, particularly for RL..."**
>
> **A1:** First, post-training compute are captured within our unified formulation. In Eq. (4),
>
> E_train ≈ (P_GPU / θ_GPU) × PUE × F_total^train × A_time × EF_region,
>
> Post-training costs enter through:
> - (1) additional optimization stages increase the cumulative training compute (F_total^train);
> - (2) iterative alignment and engineering overhead increase the runtime amplification factor (A_time).
>
> Common post-training stages, such as supervised fine-tuning and RL-based alignment, are subsumed in our aggregate compute formulation (as Appendix C). Particularly, RL-based alignment does not merely rescale pretraining FLOPs; it introduces additional compute pathways (e.g., reward evaluation). In our framework, these effects are captured in aggregate through effective compute scaling and runtime amplification.
>
> This level of abstraction is not a weakness of the framework, but a necessary design choice that enables broad applicability across heterogeneous training pipelines.
>
> Second, empirical grounding supports these estimates:
> - **Direct Capture (Tier 1):** When total energy or GPU-hours are reported, all realized training stages (including RL) are inherently included in the measurements.
> - **Statistical Inference (Tier 2 & 3):** When metadata are incomplete, we rely on regression-calibrated mappings from FLOPs to emissions, which implicitly capture the average computational overhead of real-world training pipelines. This provides a scalable empirical approximation under incomplete disclosure, rather than relying on ad hoc assumptions.
>
> The lack of detailed information about all RL runs is a systemic issue in current industry reporting practices. Therefore, our framework estimates the aggregate contribution of RL under realistic reporting constraints, rather than reconstructing specific stage-by-stage training pipelines. Our sensitivity analysis further shows that, even with minimal metadata, the aggregate estimates remain predictive at an order-of-magnitude level.
>
>
>
> > **W2: "Another hidden computational cost that was not considered in this work is hyperparameter tuning."**
>
> **A2:** We thank the reviewer for this important point. We agree that hyperparameter tuning and repeated experimental runs can introduce additional compute.
>
> First, we clarify the **accounting boundary** of our work. Our work targets the carbon emissions of individual model instances (i.e., finalized models released in public repositories), rather than the full project-level R&D lifecycle. Hyperparameter tuning is often performed at the model-family level, producing many intermediate variants. Attributing all such costs to each released model would lead to systematic double-counting across repositories. We therefore focus on the realized cost of each published instance on the Hugging Face platform.
>
> Second, although hyperparameter tuning is not an explicit input variable in our formulas, **its instance-level effects are implicitly captured**. Our estimates are defined at the level of a single final trained model. Tier 2 and Tier 3 mappings are calibrated on Tier 1 models with reported cumulative GPU-hours or energy usage, which already include restarts and tuning overhead. As a result, these costs are amortized into our empirical estimates.
>
> Finally, explicitly modeling all hyperparameter tuning is infeasible at scale due to the lack of disclosure of tuning logs and failed runs. This limitation is systematic across current reporting practices. We agree that full-lifecycle accounting, including all tuning and experimentation, is an important direction, and we leave it to future work beyond the current model-centric scope.

---

> > ### Author Rebuttal · Reviewer_wy2M · 2026-04-04
> >
> > Thanks for the response. I do not have further questions.

---

### Decision · Program_Chairs · 2026-04-30

**Decision:**

Accept (regular)

**Comment:**

This paper provides a timely and first-of-its-kind large-scale analysis of the training emissions of highly-downloaded open-source ML models on Hugging Face. While several questions were raised regarding the assumptions made in the analysis, these assumptions were either ultimately found to be defensible and/or were explicitly addressed via additional analysis in the rebuttal, with confidence that the authors would revise the manuscript accordingly to further address these assumptions and further clarify the analysis scope. Questions were also raised about the large estimation uncertainty, but reviewers appreciated the inherent difficulty of large-scale real-data analysis and the transparency with which this was addressed. Overall, this analysis is valuable and timely, and warrants acceptance to ICML.

Note: Reviewers raised a point that was not addressed by the authors in the rebuttal: FLOPS and actual energy usage are not always directly correlated. The authors should be sure to address this point in the revised version.